

# Sensitivity of active layer freezing process to snow cover in Arctic Alaska

Yonghong Yi[1]*, John S. Kimball[2], Richard H. Chen[3], Mahta Moghaddam[3], Charles E. Miller[1]

[1]Jet Propulsion Laboratory, California Institute of Technology, 4800 Oak Grove Drive, Pasadena CA, USA
[2]Numerical Terradynamic Simulation Group, The University of Montana, Missoula MT, USA
[3]Department of Electrical Engineering, University of Southern California, CA, USA

*Correspondence to*: Yonghong Yi (yonghong.yi@jpl.nasa.gov)

**Abstract.** The contribution of cold season soil respiration to Arctic-boreal carbon cycle and potential feedbacks to global climate system remain poorly quantified, partly due to a poor understanding of the changes in the soil thermal regime and liquid water content during the soil freezing process. Here, we characterized the processes controlling active layer freezing in Arctic Alaska using an integrated approach combining in-situ observations, local scale (~50 m) longwave radar retrievals from NASA Airborne P-band polarimetric SAR (PolSAR), and a remote sensing driven permafrost model. To better capture landscape variability in snow cover and its influence on soil thermal regime, we downscaled global coarse-resolution (~0.5°) reanalysis snow data using finer scale (500 m) MODIS (MODerate resolution Imaging Spectroradiometer) snow cover extent (SCE) observations. The downscaled 1-km snow depth dataset captured fine-scale variability associated with local topography, and compared well with in-situ observations across Alaska, with a mean RMSE of 0.16 m and bias of -0.01 m in Arctic Alaska, which was used to drive the permafrost model. We also used the in-situ soil dielectric constant ($\varepsilon$) profile measurements to guide model parameterization of soil organic layer and unfrozen water content curve. Across a 2° latitudinal zone along the Dalton highway in the Alaska North Slope, the model simulated mean zero-curtain period was generally consistent with in-situ observations (R: 0.6±0.2; RMSE: 19±6 days), which showed mean zero-curtain periods of 61±11 to 73±15 days from depths of 0.25 m to 0.45 m. Along the same transect, both the observed and model simulated zero-curtain periods were positively correlated (R > 0.55, p < 0.01) with snow cover fraction (SCF) derived from MODIS SCE data from September to October; this was also consistent with findings based on the airborne radar $\varepsilon$ retrievals in 2014 and 2015. The $\varepsilon$ difference of the surface soil layer between late August and early October was negatively correlated with MODIS SCF in September (R = -0.77, p < 0.01); areas with lower SCF generally showed larger $\varepsilon$ reductions, indicating earlier soil freezing. At regional scales, simulated zero-curtain period in the upper (< 0.4 m) active layer showed large variability and was closely correlated with variations in early season snow cover. Areas with earlier snow onset generally showed a longer zero-curtain period; however, the soil freeze onset and zero-curtain period in deeper (> 0.5 m) soil were more closely linked to the maximum thaw depth. With a mean soil freezing lagging rate of $0.79 \pm 0.52$ days cm[-1] at depth of 0.35 m indicated by in-situ soil temperature profile, deepening active layer associated with climate warming will lead to a longer unfrozen period in the deeper soils and potentially result in more soil carbon loss during cold season.



## 1 Introduction

Warming in the northern high latitudes is occurring at roughly twice the global rate, leading to widespread soil thawing and permafrost degradation (Liljedahl et al., 2016). Increasing soil warming and thawing potentially expose vast soil organic carbon (SOC) stocks stored in permafrost soils to mobilization and decomposition, which may promote large positive climate

feedbacks (Schuur et al., 2015). The timing, magnitude, location and form of this potential permafrost carbon feedback remain highly uncertain due to many poorly understood mechanisms that control permafrost thaw and subsequent organic carbon decomposition (Lawrence et al., 2015). Despite recent improvements in modelling permafrost soil thermal and carbon dynamics, global model estimates of near-surface permafrost loss by 2100 range from 30% to 99% and associated carbon release ranges from 37-174 Pg C under current climate warming trajectory (Representative Concentration Pathway RCP 8.5)

(Koven et al., 2013; Schuur et al., 2015). Moreover, most observational and modelling studies in the Arctic-Boreal Zone (ABZ) have focused on the shorter growing season, while cold season soil respiration may account for more than 50% of the annual carbon budget (Zona et al., 2016).

A lack of consensus on the contribution of cold season soil respiration to the annual ABZ carbon cycle and potential carbon

feedbacks of ABZ ecosystems to the global climate system can be largely attributed to relatively poor understanding of changes in liquid water content and the soil thermal regime that occur during the seasonal soil freeze/thaw (F/T) transition (Oechel et al., 1997; Zona et al., 2016). Models typically assume that the thaw or growing season is the most active period of carbon exchange in ABZ ecosystems, and soil respiration shuts down when surface soils freeze (Commane et al., 2017). However, unfrozen conditions in deeper soil layers can persist for substantially longer time than surface soils and maintain a significant

amount of liquid water, sustaining soil respiration for several weeks or more (Oechel et al., 1997). Earlier snow accumulation and a deeper snowpack can effectively insulate soils from cold air temperatures (Zhang, 2005; Yi et al., 2015). Soil moisture can further delay soil freezing due to large latent heat release with soil water phase change, where soil temperatures can persist near 0°C (i.e. the zero-curtain period) for up to several weeks or more during the late autumn and early winter seasons. This zero-curtain period can sustain soil microbial activity and has been shown to be closely correlated with soil respiration during

autumn and early winter (Zona et al., 2016; Euskirchen et al., 2017). On the other hand, highly organic soils and peat, (e.g. SOC>25 kg C m$^{-2}$), prevalent in the ABZ, can act as a strong insulator during the summer thaw season, and can also have a significant impact on the soil thermal regime and hydrologic processes due to its distinct hydraulic and thermal properties (Lawrence and Slater, 2008; Nicolsky et al., 2017).

We still lack a comprehensive understanding of how the soil freezing process and zero curtain period vary across the Arctic and are responding to recent climate trends and associated changes in snow cover conditions, especially at deep soils. Limited field studies have shown inconsistent trends in the autumn soil freeze-up and zero-curtain period in the Arctic, mainly attributed to relatively short study period and large inter-annual climate variability (Smith et al., 2016; Euskirchen et al., 2017; Kittler et




al., 2017). Moreover, sparse in-situ measurements covering different temporal periods pose great challenges in characterizing regional trends in soil freezing across the Arctic. Satellite data records over the past three decades indicate widespread reductions (~0.8-1.3 days decade[-1]) in the mean annual frozen season across the pan-Arctic domain (Kim et al., 2015). This is primarily caused by earlier spring thawing, while the onset of autumn soil freezing shows more variable trends partly due to

more variable snow cover conditions during fall and winter (Qian et al., 2011; Brown and Derksen, 2013; Burke et al., 2013). Moreover, regional monitoring of soil F/T dynamics from current satellite observations mostly rely on high-frequency passive microwave sensors and scatterometers with relatively coarse spatial resolution (~≥ 10 km) and limited sensitivity to deep (~< 5 cm) soils (Kim et al., 2015; Derksen et al., 2017).

Detailed process models have been widely used to simulate soil F/T and permafrost dynamics in the ABZ, which can well represent heat transfer between the atmosphere and underlying soil and permafrost layers to predict changes in active layer conditions, and land-atmosphere interactions (Burke et al., 2013; Rawlins et al., 2013; Lawrence et al., 2015; Paquin and Sushama, 2015; Jafarov and Schaefer 2016). However, their applications are constrained by large uncertainties in surface meteorology drivers, deficient representations of surface heterogeneity and microtopography, and insufficient understanding

of the processes controlling soil F/T and permafrost dynamics (e.g. Koven et al., 2013; Slater and Lawrence, 2013; Walvoord et al., 2016). Other models provide an intermediate level of complexity by relying on a simplified process logic utilizing satellite remote sensing based environmental observations as key model drivers; these models have been effective in regional scale mapping of permafrost extent and active layer dynamics in the Arctic (Park et al., 2016; Westermann et al., 2017; Yi et al., 2018).

We developed a remote sensing driven permafrost model and successfully applied it to simulate the soil active layer dynamics across Alaska at 1km resolution (Yi et al., 2018). Seasonal snow cover insulation is one of the most important factors influencing soil freezing, while few high-resolution (≤1 km) spatial snow datasets are available for the Arctic region. In the previous study (Yi et al., 2018), we mainly used coarse-resolution (~0.5°) global reanalysis snow data including snow water

equivalent, snow depth and snow cover fraction to drive the model. The snow water equivalent and snow depth data were used to calculate snow density and snow thermal properties. However, in this study, we developed a new algorithm to downscale the coarse-resolution reanalysis data using finer spatial resolution satellite snow cover extent (SCE) observations from the MODerate resolution Imaging Spectroradiometer (MODIS) sensor along with other supporting ancillary data. The permafrost model was then driven using the downscaled 1-km snow datasets and other remote sensing records to simulate the active layer

freezing process, including soil freeze onset and the zero-curtain period across the Alaska Arctic. We also investigated the sensitivity of soil dielectric constant from both in-situ measurements and airborne radar retrievals derived from NASA Airborne P-band polarimetric SAR (PolSAR) backscatters to active layer freezing process, which help constrain the model parameterization and evaluate model simulations. Finally, we analysed the regional sensitivity of model simulated soil freezing process to recent climate variability (2001-present).



## 2 Methods

### 2.1 Model description

We developed a remote sensing driven permafrost model, which was used to simulate the permafrost distribution and active layer dynamics in Alaska at 1-km resolution (Yi et al., 2018). The model simulations were conducted using a detailed soil

process model (Rawlins et al., 2013; Yi et al., 2015) primarily driven by global satellite observations of key parameters including land surface "skin" temperature (LST), SCE and surface to root zone (≤ 1 m depth) soil moisture (SM). The soil model accounts for effects of soil organic layer, changes in surface snow cover properties and soil water phase change on the soil freeze/thaw (F/T) process, which are important controls on active layer thermal dynamics in the ABZ (Nicolsky et al., 2007; Lawrence and Slater, 2008; Yi et al., 2015). The model can simulate the soil F/T process and temperature profile down

to 60 m below surface using 23 soil layers, with increasing layer thickness at depth (soil nodes from 0-1m: 0.01, 0.03, 0.08, 0.13, 0.23 ,0.33, 0.45, 0.55, 0.70, 1.05 m). Up to five snow layers are used to account for the effects of seasonal snow cover evolution and changes in seasonal snow density and thermal properties are also considered. Both snow heat capacity and thermal conductivity vary with snow density, and empirical methods are used to estimate these two variables based on snow density (Calonne et al., 2011).

The soil model simulates snow and ground thermal dynamics by solving a 1-D heat transfer equation with phase change (Nicolsky et al., 2007; Rawlins et al., 2013):

$$C\frac{\partial}{\partial t}T(z,t) + L\zeta\frac{\partial}{\partial t}\theta(T,z) = \frac{\partial}{\partial z}\left(\lambda\frac{\partial}{\partial z}T(z,t)\right),$$
$$z \in [z_s, z_b]$$

(1)

where $T(z,t)$ is the temperature (°C), $C$ and λ are the volumetric heat capacity (J m$^{-3}$ K$^{-1}$) and thermal conductivity (W m$^{-1}$ K$^{-1}$)

$^{-1}$) of soil respectively, varying with soil moisture, freeze/thaw state and depth; $L$ is the volumetric latent heat of fusion of water (J m$^{-3}$); $\zeta$ is the volumetric water content, and $\theta$ is the unfrozen liquid water fraction (range: 0-1). The upper boundary condition is defined as the surface temperature at the snow/ground surface (i.e. LST), while a heat flux charactering the geothermal gradient is applied at the lower boundary. The unfrozen liquid water fraction ($\theta$) is estimated empirically as:

$$\theta = \begin{cases} 1 & T \geq T_* \\ |T_*|^b |T|^{-b} & T < T_* \end{cases}$$

(2)

Soil water usually freezes at a sub-zero temperature depending on solute concentration and other factors, and the constant $T_*$ is used to represent this freezing point depression with values generally well above -1°C (Banin and Anderson, 1974; Woo, 2012). $b$ is a dimensionless parameter determined by fitting the unfrozen water curve (Romanovsky and Osterkamp, 2000;



Schaefer and Jafarov, 2016). A significant amount of liquid water can exist even when the soil temperature is considerably lower than $T_*$, which is characterized by different values of $b$. Fine-grained soils can have a larger amount of liquid water below freezing and thus are generally associated with smaller $b$ values (Woo, 2012).

## 2.2 Constructing a high-resolution snow dataset

Most regional and global permafrost models rely on relatively coarse precipitation or snow inputs obtained from global reanalysis data to represent snow insulation effects on the soil thermal regime. However, coarse-resolution reanalysis datasets generally have difficulty capturing landscape-scale (100-1000 m) variability in snow cover conditions, especially over complex terrain and during seasonal transitions (Liston and Sturm, 2002; Gisnas et al., 2016). We previously used a simple methodology to generate a continuous 1-km 8-day snow depth and density records for estimating active layer dynamics by combining

MODIS 500 m SCE data with coarser (~0.5°) snow depth records with the MERRA-2 global reanalysis (Yi et al., 2018). We first interpolated the MERRA-2 data over a finer 1-km spatial grid using an inverse-distance weighting scheme, and then used the MODIS SCE product to identify snow-free pixels within each 0.5° MERRA-2 grid cell and adjust the 1-km snow depth estimates accordingly. However, a more sophisticated downscaling scheme was developed for this investigation to better account for the influence of local topography on the 1km snow distribution pattern. The MODIS record provides some

information on the SCE distribution, but is constrained by data loss from persistent cloud cover, which can account for more than 10% of the Alaskan domain during the transition season at 8-day temporal scale (Fig. S1). To overcome this problem, we developed an elevation-based spatial filter algorithm to predict the snow occurrence for the MODIS cloud contaminated pixels; we then used the gap-filled MODIS data to downscale the MERRA-2 snow depth data.

### 2.2.1 Cloud filtering of MODIS SCE data

Most existing cloud filter algorithms designed for the MODIS SCE products use empirical relationships between snow cover conditions and ancillary data to predict snow cover occurrence for cloud-covered pixels (e.g. Parajka and Bloschl, 2008; Gafurov and Bardossy, 2009; Parajka et al., 2010). These algorithms are generally appropriate for the limited areas and conditions in which they were developed and may not be adaptable for other regions with different climate or topography. To develop a more general cloud filter algorithm, we borrowed from spatial interpolation methods designed for generating grid-

based surface meteorology from in situ weather station observations. We used a similar methodology that was used to generate Daymet surface precipitation, which uses a truncated Gaussian weighting filter and accounts for the dependence of precipitation on elevation (Thornton et al., 1997). This method was found to generate reliable precipitation estimates in complex topography in the western US (Henn et al., 2018). For our application, we treated the pixels without cloud cover as "station observations", and then used the spatial filter to predict the occurrence of snow in cloud-contaminated pixels for

generating continuous cloud-free snow cover images at 1km spatial resolution and 8-day time scale.

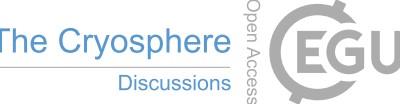

The general form of the spatial filter, with respect to the cloud contaminated or central pixel ($i$) to be filled, is defined as:

$$W(d) = \begin{cases} 0; & if\ d \geq R \\ \exp\left[-\alpha\left(1-\left(\frac{d}{R}\right)^2\right)\right]; & if\ d < R \end{cases} \qquad (3)$$

Where $W(d)$ is the filter weight associated with the radial distance d from the central pixel, and $\alpha$ is a unitless shape parameter with a prescribed value of 6.0 following Thoronton et al. (1997). $R$ is the truncation distance, varying with the local density of "observations" (i.e. cloud-free pixels) in the adjacent areas of the central pixel; at least 50 "observations" should be included for interpolating to the central pixel, with a maximum search radius of 50 km. Snow distribution is closely associated with local topography, therefore we divided the "observations" falling within the range of the search radius into two groups: elevations above and below the elevation of the central pixel. We then estimated the snow occurrence probability ($P_{snow}$) and weighted elevation ($Z$) for each group:

$$P_{snow} = \frac{\sum_{j=1}^{n} W(d_j) \times P_j}{\sum_{j=1}^{n} W(d_j)} \qquad where\ P_j = \begin{cases} 1; & if\ snow\ exists \\ 0; & if\ snow\ does\ not\ exist \end{cases}$$

$$Z = \frac{\sum_{j=1}^{n} W(d_j) \times Z_j}{\sum_{j=1}^{n} W(d_j)} \qquad (4)$$

Then the snow occurrence probability at the central pixel ($P_{snow,i}$) was estimated as a weighted function of the snow occurrence probability of the two groups ($Z_{above}$ and $Z_{below}$):

$$P_{snow,i} = P_{snow,below} + (P_{snow,above} - P_{snow,below}) \times (Z_i - Z_{below})/(Z_{above} - Z_{below}) \qquad (5)$$

The snow cover condition (SC) of the central pixel is determined based on the comparison of $P_{snow,i}$ with a specific cutoff value, $P_{cutoff}$:

$$SC = \begin{cases} 0; & P_{snow,i} < P_{cutoff} \\ 1; & P_{snow,i} \geq P_{cutoff} \end{cases} \qquad (6)$$

A simple temporal filtering of the MODIS SCE dataset was conducted prior to the application of the spatial filter. Pixels with cloud cover were reclassified as either snow or non-snow conditions if the two temporally adjacent periods were both identified as cloud free and indicated consistent snow or non-snow covered conditions. Missing SCE pixels occurring during polar night were assigned as "snow" when there were established snow cover conditions in the prior 8-day period or there was more than 0.2 m snow depth indicated in the co-located MERRA-2 grid cell. This procedure effectively reduced the number of cloud contaminated pixels requiring spatial gap-filling.

### 2.2.2 Downscaling of MERRA-2 snow depth data

The resulting gap-filled 8-day MODIS SCE data was used with a 1-km digital elevation model (DEM) aggregated from the 2 arc-second (~ 60 m) DEM map for Alaska (USGS, 2017) to downscale the MERRA-2 snow depth data to 1-km resolution. Here, a spatial filter similar to the above procedure was used for the downscaling process, except that the MERRA-2 gridded snow data were treated as the station "observations" and the "station" elevations were defined as the mean elevation within





the associated ~0.5° MERRA-2 grid cell. Previous studies have demonstrated a clear dependence of snow depth on elevation, generally with snow depth increase with elevation up to a certain level followed by a decrease at the highest elevations (Grünewald et al., 2014; Kirchner et al., 2014). Therefore, we used the transformed snow depth variables instead of the original MERRA-2 data as inputs to the spatial filter to account for the spatial dependence of snow distribution on elevation. We used

least-square regression to analyse the relationship between snow depth and elevation:

$$\left(\frac{SD-SD_{min}}{SD_{max}-SD_{min}}\right) = \beta_0 + \beta_1\left(\frac{Z-Z_{min}}{Z_{max}-Z_{min}}\right) \qquad (7)$$

The snow depth (SD) and elevation (Z) data were normalized using maximum and minimum values from the MERRA-2 grid cells within the spatial search radius to account for local variability in the snow distribution (Grünewald et al., 2014). Linear regression does not account for the snow depth decrease at the highest elevations; however, the coarse MERRA-2 data

represent average conditions within each ~0.5° grid cell and will not be able to capture snow depth changes at these high elevation extremes.

For each 1-km snow covered pixel indicated by MODIS SCE data, snow depth is estimated as:

$$SD_i = \frac{\sum_{j=1}^{n} W(d_j) \times SC \times \left(SD_j + \left(\beta_1 \times \left(\frac{Z_i - Z_j}{Z_{max}-Z_{min}}\right)\right) \times (SD_{max}-SD_{min})\right)}{\sum_{j=1}^{n} W(d_j) \times SC} \qquad (8)$$

Where the interpolation only weights MERRA-2 grid cells with snow occurrence (indicated by SC). The MERRA-2 snow depth record was used directly for the spatial interpolation (i.e. $\beta_1 \cong 0$) where no significant relationship was indicated between elevation and snow depth changes within the search radius.

### 2.3 Datasets

### 2.3.1 Model inputs

Model simulations were conducted over the Arctic Alaskan domain (>66.55° N, Fig. 1), encompassing an area of ~400,000 km$^2$. Primary model drivers included MODIS 8-day composite 1-km LST (MOD11A2; Wan et al., 2015) and 500-m SCE records (MOD10A2; Hall and Riggs, 2016), SMAP 9-km NatureRun (Version 4) and Level 4 daily surface (≤ 5 cm depth) and root zone (0-1 m depth) soil moisture (L4SM, Reichle et al., 2017), and daily snow depth and snow density from MERRA-2 global (~ 0.5° resolution) reanalysis data (Gelaro et al., 2017). The MODIS LST and SMAP L4SM products were used to

define model boundary conditions and soil thermal properties. The soil process model was run at 1-km resolution and 8-day time step consistent with the MODIS LST and SCE inputs. All model input datasets were reprojected to a consistent 1-km Albers projection prior to the model simulations. Snow depth inputs to the model were derived at a 1km resolution and 8-day time step using the MODIS SCE record and DEM downscaled from MERRA-2 snow depth records as described in Section 2.2. Compared with snow depth, snow density shows much smaller temporal and spatial variability (Sturm et al., 2010);





therefore, we still used the 1-km snow density data generated using a simple spatial interpolation scheme (Yi et al., 2018). Other ancillary inputs to the soil model included the 30-m National Land Cover Database (NLCD 2011; Jin et al., 2013), 2 arc-second (~ 60 m) DEM for Alaska (USGS, 2017), 50-m SOC estimates for Alaska (to 1-m depth; Mishra et al., 2016), and the global 9-km mineral soil texture data developed for the SMAP L4SM algorithm (De Lannoy et al., 2014). The dominant

NLCD land cover type within each 1-km pixel was used to define the modelling domain, with open water and perennial ice/snow areas excluded from the model simulations. The soil texture and SOC data were used to define model soil properties including thermal conductivities and heat capacities. The SOC inventory data was distributed through the top 10 model soil layers (≤ 1.05 m depth) following an exponentially decreasing curve (Hossain et al., 2015) to calculate the soil carbon fraction of each soil layer, and adjust the soil physical properties of each soil layer based on the weighted mineral and organic soil

components. More details on the data processing can be found in Yi et al. (2018).

### 2.3.2 In-situ data

A variety of in-situ measurements were used for model calibration and validation (Table 1). These data included half-hourly soil dielectric constant (ε) and temperature profile measurements from a Soil moisture Sensing Controller and oPtimal Estimator (SoilSCAPE) site (Moghaddam et al., 2010); daily soil temperature profile measurements from Global Terrestrial

Network for Permafrost (GTN-P) sites (Biskaborn et al., 2015), and active layer thickness (ALT) measurements from regional Circumpolar Active Layer Monitoring (CALM) network (Brown et al., 2000) sites (Fig. 1). The SoilSCAPE soil temperature and ε measurements were obtained from 4 different depths (0.05, 0.15, 0.35, 0.56 m) at 4 different sensor nodes of a wireless sensor network deployed near Prudhoe Bay, Alaska (70°13'47"N, 148°25'19"W) in the summer of 2016. Soil dielectric properties are strongly correlated with soil moisture, texture and freeze/thaw state (Mironov et al., 2010); and ε can capture

the soil freezing process well due to large differences in ε between liquid water and ice (Dobson et al., 1985), especially during the zero-curtain period, when soil temperatures hover around 0 °C (Fig. 2). The SoilSCAPE measurements were used to calibrate the model unfrozen water content curve parameterization [Eq. 2] assuming a linear relationship between ε and liquid water content (Mironov et al., 2010; Park et al., 2017). However, the slope of this linear relationship may change during the freezing period because of different dielectric properties of free and bound water, and ice (Mironov et al., 2010). The ε

measurements were also used to determine the timing of complete soil freeze-up at different depths at this site. The timing of soil freeze-up was defined when the observed ε drops below a critical level:

$$\varepsilon < \delta \times (\varepsilon_{max} - \varepsilon_{min}) \tag{9}$$

Where $\varepsilon_{max}$ and $\varepsilon_{min}$ are the maximum and minimum dielectric constant for each soil layer respectively, and the threshold $\delta$ is an empirical parameter.

The above threshold was also used to determine the critical value of the unfrozen water content and soil temperature at soil freeze-up, which were used to determine the soil freeze onset from both the GTN-P soil temperature measurements and model





simulations. For the model simulations at the SoilSCAPE site, we also used the in-situ ε profile to guide model parameterization of soil thermal properties. In the ABZ landscape, soil thermal properties were largely determined by the soil organic fraction (Lawrence and Slater, 2008). During the thaw season, the ε values were mainly determined by the soil texture and saturation degree. Since the soil is mostly saturated at this site, much larger ε values in the top two layers (Fig. 2) should

be related to organic-rich soils with large soil porosity (thus high volumetric soil moisture). Therefore, we set the first 5 soil layers of the model (0-0.23 m) in the model as organic soils, and adjusted the model thermal properties accordingly. For the GTN-P in-situ measurements, we only selected the sites where shallow ground temperature measurements (generally down to 1m depth) were available for at least two consecutive years. Most GTN-P sites meeting these criteria in Arctic Alaska are located along the Dalton Highway (Fig. 1 & Table S1); airborne P-band radar data was also acquired along this transect.

Therefore, we focused our analysis mostly on this transect. In addition, daily snow depth measurements using ultrasonic sensor were available at SNOTEL (SNOwpack TELemetry) sites across Alaska (Schaefer and Paetzold, 2000; http://www.wcc.nrcs.usda.gov, Fig. S2) and used to validate the 1-km snow depth product (Section 2.2).

### 2.3.3 Airborne radar retrievals

Soil dielectric constant retrievals from airborne radar flight transects in Arctic Alaska (Fig. 1) were used to analyse relations

between the active layer freezing and seasonal snow cover. Multiple flight lines were acquired in late August (fully thawed) and early October (partially frozen) during year 2014 and 2015 in preparation for the NASA Arctic Boreal Vulnerability Experiment (ABoVE) campaign, while our analysis largely focused on the Dalton Highway flight transect (DHN, 148.39-149.05°W, 68.78-70.40°N). Here, the airborne land parameter retrievals were derived using time series information from NASA Airborne P-band (430 MHz) PolSAR radar backscatter measurements acquired in August and October (Chen et al., in

review); this differed from our previous study which used combined (L+P-band) radar backscatter observations from NASA UAVSAR (Uninhabited Aerial Vehicle Synthetic Aperture Radar) and P-band PolSAR sensors for the active layer retrievals (Yi et al., 2018). The alternative single channel (P-band) time series algorithm used in the current study avoids potential artifacts introduced from variable acquisition times between the P and L-band sensors during the ABoVE airborne campaign in 2017. The time series approach uses the same three-layer soil dielectric model to account for differences in soil dielectric

properties between the surface organic layer, deeper active layer and underlying permafrost. In August, the three layers represent the surface thawed layer, middle and bottom active layer, and frozen layer (i.e. permafrost), while in October, the surface two layers represent a partially frozen active layer with a frozen surface layer overlying a deeper unfrozen layer. The thawed portion of the active layer from the early October flight was assumed to have the same depth to permafrost as the fully thawed active layer from late August flight. An iterative model inversion scheme was used to estimate multiple active layer

parameters by minimizing differences between the observed radar backscatter measurements and forward radar scattering model simulations. The radar retrievals include the thickness and soil dielectric constant of the surface layer in both August and October, depth and dielectric constant of deep active layer. The difference in the soil dielectric constant retrievals of the surface soil layer between August and October mostly represents the difference of liquid water content and thus indicate the

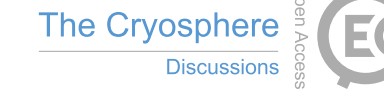

frozen conditions of surface soils. Initial validation indicated that the radar retrieved ALT at the Dalton highway transect show comparable accuracy with the in-situ measurements from the CALM sites (Chen et al., 2017; Chen et al., in review).

## 2.4 Data analysis

The soil process model was spun-up for 50 years to bring the top 10-m soil temperature profile into dynamic equilibrium with
model inputs for the year 2000 (Yi et al., 2018), followed by a model transit run from 2001 to 2017. Unfrozen conditions in the deep portion of the active layer may persist well into the winter period and into the subsequent calendar year. In order to accurately estimate the freeze onset and zero-curtain period of the active layer for the current year, model simulations of the next calendar year were also needed. Therefore, the soil freeze onset and zero-curtain period in year 2017 were not calculated. The soil freeze onset for each layer was determined when the model simulated unfrozen water content dropped below a critical
threshold indicated from in-situ soil dielectric constant measurements and the linear relationship between unfrozen water content and soil dielectric constant (Section 2.3.2). The land surface freeze onset was defined when the mean MODIS LST during 3 consecutive 8-day periods dropped below 0 °C. The zero-curtain period at each soil depth was defined as the duration between land surface freeze onset and soil freeze onset of that layer. The regional correlation between snow onset calculated from MODIS SCE data and the zero-curtain period for each soil layer was used to investigate relations between the timing of
early snow accumulation and soil freeze-up. The timing of snow onset was chosen as the centre of the 8-day period with more than 3 adjacent snow-covered periods within a 40-day moving window; the relatively long temporal window was used to account for more variable snow cover conditions during fall.

We also conducted an integrated analysis of in-situ ground observations, model simulations and airborne radar retrievals to
investigate the sensitivity of soil freezing to snow cover conditions, mostly focusing on the DHN flight transect (Fig. 1). Because the MODIS 8-day SCE product (i.e. MOD10A2) only provides binary snow data (i.e. snow vs non-snow), the dataset was binned for each 0.1° latitudinal region along the radar flight transects to calculate the area fraction of snow cover extent (SCF). Similarly, we also averaged the airborne radar retrieved surface soil dielectric constant for each 0.1° latitudinal bin. Across the radar transect, we analysed the correlations between SCF and the variables representing the soil freezing process,
which are the surface dielectric constant changes derived from the airborne data, and zero-curtain period derived from soil model simulations and in-situ observations (only for DHN transect). For the site analysis, the SCF from the 0.1° latitudinal bin including the site was used. The Arctic Alaska is generally fully covered by snow by the end of October or early November. Therefore, for the zero-curtain period analysis, we used the SCF averaged from September to October, while only SCF in September was used for the airborne radar data analysis since radar data was obtained in early October.



## 3. Results

### 3.1 Model validation

Accurate simulations of early cold season soil freezing require accurate characterization of landscape-scale snow cover conditions, which in turn requires gap-filling MODIS SCE products to account for the pervasive cloud cover in the northern

high latitudes (Section 2.2). The gap-filled MODIS SCE products were then combined with other ancillary data to downscale the MERRA-2 reanalysis snow depth data, as one of the main driver datasets for the permafrost model. The gap-filled MODIS SCE product were first cross-checked using the two MODIS sensors (Terra and Aqua); the downscaled snow depth data were evaluated using in-situ SNOTEL observations across Alaska. The model simulated soil freezing process and ALT were evaluated using multiple in-situ datasets across Arctic Alaska.

### 3.1.1 Regional 1-km snow cover product

The cloud-free Aqua MODIS SCE data were used to evaluate the accuracy of filled pixels identified as cloud covered in the Terra MODIS SCE data and vice versa, assuming relatively consistent snow conditions in the morning (Terra) and afternoon (Aqua) images at the 8-day time scale. Our results indicate an accuracy of more than 80% in cloud filtering algorithm without obvious differences observed between the two sensors (Table 2 & Fig. S1). The cloud cover fraction for the 8-day temporal

composite Terra MODIS SCE data ranges from 0.5% to 10.1% of the entire state throughout the year, and the percentage of cloud-free Aqua MODIS pixels that overlapped with cloud-covered Terra MODIS pixels ranges from 0.4% to 4.6% (Fig. S1). There are significantly more cloud-covered pixels in Aqua MODIS (1.1%-15.2%), and thus more cloud-free Terra MODIS pixels (0.9% to 9.8%) are overlapped with cloud-covered Aqua MODIS pixels. Cloud cover mostly occurs in the spring and fall shoulder seasons, resulting in larger SCE uncertainties during those periods. There is no obvious bias in the

misclassification of cloud-contaminated pixels (Table 2), which indicates that a cut-off threshold of 50% works well for detecting snow occurrence (Eq. 6; Table 1). Using a higher threshold (e.g. 60%) generally results in more snow pixels misclassified as land pixels and vice versa.

Comparing with in-situ measurements from the Alaskan SNOTEL sites, the 1-km MERRA-2 snow depth data generated using

the new downscaling algorithm showed an overall improvement over the original spatial interpolation scheme used in Yi et al. (2018) (Table 3). The new 1-km snow depth data showed overall reduced RMSE values and lower biases except in Interior Alaska at elevations between 400 m and 800 m. At elevations between 400 m and 600 m, the USGS DEM data used for the downscaling scheme at the 1-km grid encompassing the SNOTEL sites shows much large deviation from the in-situ elevation reported by the SNOTEL dataset (Fig. S3), which likely explains relatively poorer performance of the new 1-km snow depth

dataset in this elevation band. In Arctic Alaska, our new snow depth product modestly improves over the Yi et al. (2018) product, with RMSE of 0.16 m and bias of -0.01 m, versus RMSE of 0.18 m and bias of -0.03 m for the original dataset. However, there are only eight SNOTEL sites in this region, and only two sites on North Slope Alaska. Comparing with the Yi



et al. (2018) interpolated product, our new MERRA-2 down-scaled snow product captured more fine-scale details of spring snow melting and topographically varying winter snow distribution, especially in mountain areas (Fig. 3 & Fig. S4).

The snow offset and onset derived from the MODIS SCE data and downscaled MERRA-2 snow depth data show very similar spatial patterns and temporal trends from 2001 to 2016 (Fig. S5 & S6). These results indicate that the downscaled MERRA-2 snow depth data generally captures the regional variability in snow cover conditions during the transitional season indicated from the MODIS observations. During the study period, both datasets show similar spring snow offset dates with DOY 138±7 for MODIS versus DOY 140±7 for the downscaled snow depth data, while there is an approximate 10-day difference in fall snow onset dates between the two datasets (MODIS: DOY 284±5; MERRA-2: DOY 273±5). The difference in mean snow onset is most likely due to the different methods used to determine snow onset for the two datasets. For the MODIS SCE record, snow onset was chosen as the composite period with more than 3 adjacent 8-day snow covered periods within a 40-day moving window, while snow onset was defined from the downscaled snow depth record as the composite period with mean snow depth above a 0.05 m threshold within a 24-day moving window. A higher snow depth threshold results in a later snow onset date in the MERRA-2 dataset. However, the snow onset derived from these two datasets show similar spatial pattern (Fig. S5), and interannual variability during the study period (R=0.79, p<0.01). For both datasets, Arctic Alaska showing primarily earlier snow onset during the study period, which will be discussed in Section 3.2.

### 3.1.2 Soil model simulations

We tested different soil dielectric thresholds (Eq. 9) to determine the soil freeze-up and zero-curtain period at the SoilSCAPE site near Prudhoe Bay. The resulting threshold with the best fit to the in situ measurements was used to determine the critical threshold of unfrozen water content and soil temperature at soil freeze-up for both the GTN-P measurements and the soil process model simulations. Model simulated soil temperature profile was very sensitive to soil thermal conductivity, with large differences between organic and mineral soils. Therefore, we also used the soil dielectric measurements to determine the depth of surface organic layer (0-0.23 m) and adjust the soil thermal conductivity accordingly (Section 2.3.2). The soil thermal conductivity within the organic layer was assumed to gradually increase with depth to account for the increases in the soil bulk density and decomposed state (Letts et al., 2000; Fig. S7). Using this soil thermal conductivity profile, the model simulated temperatures agree well with the in-situ observations (R>0.97, RMSE<2.24 °C for all depths).

A threshold of 15% resulted in the best match of the zero-curtain period determined using in-situ soil dielectric constant measurements and model simulated unfrozen water content (Fig. 4 & Table S2), with a mean RMSE of 10.3 days averaged from depths at 0.15 m to 0.56 m in 2016 and 2017. The soil dielectric constant measurements at surface soil layers (0.05 and 0.15m) showed large temporal variability (Fig. 2) and dropped rapidly after surface freezing, resulting a much shorter duration of zero-curtain period at these two surface layers, comparing with the two deeper layers. Therefore, the short zero-curtain period of the two surface layers may not be captured by the model simulations at 8-day time step. The model estimated freeze





onset at the land surface was determined from MODIS LST records extracted at the SoilSCAPE site because no in situ surface air temperature measurements were available; therefore, the model errors for the soil freeze onset and the simulated zero-curtain period at different depths are consistent. The observed changes in the normalized soil dielectric constant with soil freezing is presented for a selected SoilSCAPE sensor node (S6) in Fig. 4 (b). The soil dielectric constant below the freezing

point and above -10°C ranges from ~5% to 20% of the soil dielectric constant of unfrozen soils. Assuming a linear relationship between the soil dielectric constant and liquid water content (Mironov et al., 2010), Fig. 4(b) can approximately represent the changes in unfrozen water content during soil freezing process. Assuming soil freeze-up starts when the soil dielectric constant drops below 15%-20% of the annual amplitude, the corresponding soil temperatures during this period range from -0.01 to -1°C at depths of 0.05, 0.15 and 0.56 m. We selected a temperature threshold of -0.35°C for soil freeze-up, which is at the higher

end of the range indicated from the S6 node, but closer to the other SoilSCAPE nodes having much more rapid changes in the soil dielectric constant below 0°C (not shown). This temperature threshold is also consistent with model simulations, which show a -0.3 to -0.5°C temperature range and corresponding to 15% to 20% of liquid water content during freeze-up, assuming a linear relationship between liquid water content and soil dielectric constant. This temperature threshold was used to determine soil freeze onset and the zero-curtain period at the GTN-P sites, where only soil temperature measurements are available. We

note, however, that in-situ soil dielectric measurements in frozen soils have significant uncertainties that may propagate to errors in the unfrozen water content, freeze-up threshold and zero curtain estimates.

Across the DHN transect, the soil process model simulated zero-curtain period was significantly ($p<0.1$) correlated with the in-situ observations, with a mean bias of 6.6 days and RMSE of 19.0 days at 0.35 m soil depth (Table 4). However, lower

correspondence was found between the model simulations and in situ observations at the DHN Happy Valley site (R=0.48, $p>0.1$). Relatively large RMSE differences in the estimated zero curtain period were mainly due to large interannual variability in the soil freeze onset and zero-curtain period during the study period (Fig. 5a). Both model simulated and in-situ observed zero-curtain periods were strongly correlated at the different soil depths, e.g. R>0.92 for the zero-curtain at 0.25 m and 0.35 m except for the SagMAT site (R=0.87). Larger differences were observed in the model simulated and in-situ observed zero-

curtain period at the surface ($\leq 0.15$ m) and for deeper soil layers ($\geq 0.45$ m) due to model limitations in capturing a shorter zero-curtain period in surface soils at the 8-day temporal scale, and larger uncertainties in both the in-situ observations and model simulated zero-curtain period in deeper soils. Across the DHN transect, both the model simulated and in-situ observed soil freeze onset during the study period were strongly correlated (R>0.9) with the zero-curtain period at soil depths below 0.15 m (Fig. 5a, Figs. S8-S9); therefore, the modelled soil freeze onset was not discussed separately. In addition, consistent

inter-annual variability in the soil freeze onset and zero-curtain period derived from in-situ observations was observed across the GTN-P sites at the DHN transect covering a ~2° latitudinal gradient (Table S1), especially for the sites located north of the Sagwon site (> 69.43°N). This will be discussed further in the next section.




Across Arctic Alaska, the model simulations slightly overestimate ALT compared with the in-situ ALT measurements from total 32 CALM sites (Fig. S10), with a mean bias of 10.0±13.2 cm (~20% of mean ALT), and RMSE of 15.6±7.7 cm. For the 23 CALM sites with at least 9 years of ALT measurements, the correlations between model simulated ALT and the in-situ measurements range from 0.20 to 0.69, with 17 sites showing a significant correlation (p<0.1).

## 3.2 Sensitivity of active layer freezing process

We first analysed the sensitivity of active layer freezing process to seasonal snow cover using in-situ observation, local-scale airborne P-band radar retrievals and model simulations, focusing on the Dalton highway transect in the Alaska North Slope. We then discussed the regional sensitivity of model simulated soil freeze onset and zero-curtain period within the active layer to early snow accumulation indicated by MODIS SCE data across Arctic Alaska.

### 3.2.1 Along the airborne radar flight transect

For all sites along the DHN transect, both the model simulated and in-situ observed zero-curtain periods at 0.25 m and 0.35 m soil depths showed significant positive correlations (R=0.69±0.14, p<0.1) with MODIS SCF. The zero-curtain period and SCF showed similar inter-annual variability across the DHN transect, particularly north of the Sagwon sites (>69.4 °N). Thus years with greater (less) snow cover (SCF) are associated with a generally longer (shorter) zero-curtain period at these depths. The average zero-curtain period at 0.35 m soil depth for the DHN transect sites located from 70.4 to 69.4 °N is shown along with the corresponding MODIS SCF observations in Fig. 5b. These observations show large inter-annual variability in early season snow accumulation in this area, with the fall (Sep-Oct) SCF varying from 0.35 to 0.77 from 2001 to 2015. Correspondingly, both the in-situ and model simulated zero-curtain period show large variability throughout the period, varying from 23.5 days to 79.2 days at 0.25 m and from 24.7 to 90.3 days at 0.35 m for the in-situ data, and varying from 29.3 to 76.0 days at 0.25 m and from 44.0 to 90.7 days at 0.35 m for the model simulations. It should be noted that the in-situ sites used for averages have different temporal coverage. However, even though early snow accumulation was the primary factor affecting the freezing process of the top soils, ALT is more closely related to the length of zero-curtain period of deeper soils, particularly in areas with a shallower thaw depth. This can be seen from delayed soil freezing below 0.30 m soil depth at two of the monitoring sites (West dock and Galbraith Lake) in years with larger ALT (Fig. 5 c-d). Both monitoring sites show deeper ALT conditions during the later years of the study period, which are also associated with larger soil freezing lag rates. Here, the soil freezing lag rate is defined as the ratio of soil freeze onset difference between two adjacent soil layers to depth difference of the two layers. However, soil freezing lag rates derived from both the in-situ measurements and model simulations show large variability and are likely associated with large uncertainty especially at deep soil layers.

The airborne P-band radar retrievals over the DHN flight transect in early October (2014: 10-09, 2015: 10-01) showed a larger reduction in the surface soil dielectric constant (ε1) in areas with shallower snow cover during September, comparing with





areas with deeper snow cover (Fig. 6). In both years, the ε1 differences were negatively correlated with MODIS SCF across the 0.1° binned latitudinal gradient along the transect (2014: R=-0.69, 2015: R=-0.76, p<0.01, n=17). The reduction in ε1 retrievals in early October was more obvious in the northern part (>69.5 °N) of the transect (2014: -10.35±5.46, 2015: -6.64±1.94) which had a much shallower seasonal snow cover (mean SCF of 0.22 in 2014 and 0.38 in 2015) than other areas,

indicating an early frozen condition in those areas. On the other hand, variations in vegetation cover may also contribute to the large contrast in the surface soil dielectric constant retrievals between the northern and southern portions of the transect. However, large changes in the soil dielectric constant retrievals between 2014 and 2015 in the northern portion of transect indicate soil dielectric constant being a relatively robust indicator of frozen soil conditions. A similar relationship between radar dielectric constant changes and MODIS SCF was observed over the Atqasuk (ATQ) flight transect. However, the

opposite pattern was observed over the Ivotuk (IVO) transect, where ε1 showed consistent changes with seasonal snow cover in accordance with altitudinal changes along the transect (Fig. S11). Here, the IVO transect includes more variable topography (Fig. 1) and higher elevations (614±75 m) than the DHN (208±36 m) and ATQ (34±7 m) transects. Additional P-band SAR flight acquisitions were also obtained over portions of western Alaska in 2015 (Fig. S12a). When we examined the regional-wide radar flight transects, we found that the ε1 difference between August and October was much larger over areas with

continuous permafrost (Fig. S12b). These results indicate earlier onset of frozen soil conditions in the colder and more continuous permafrost areas, which is consistent with the available ground truth observations and indicates that the radar retrieval algorithm is generally effective for surface soil conditions in tundra.

**3.2.2 Arctic Alaska**

We compared the model simulated zero-curtain period over Arctic Alaska for two selected years with relatively late (2007)

and early (2015) snowfall. These results indicate that snow accumulation during the early cold season is the primary control of zero-curtain period within the upper (< 0.4 m) soil layers (Fig. 7 & Fig. S13). The regional mean surface freeze onset and snow onset based on the MODIS LST and SCE observations was DOY 269±5 (surface freeze onset) and 281±9 (snow onset) in 2007, and DOY 259±8 (surface freeze onset) and 262±12 (snow onset) in 2015. The later snow cover establishment in 2007 resulted in an overall shorter zero-curtain period over most of the Arctic Alaska region, with a model simulated mean zero-

curtain period of 49.3±25.1 days at 0.25 m soil depth and 64.3±26.3 days at 0.35 m depth. In contrast, earlier snow accumulation in 2015 resulted in a longer zero-curtain period, ranging from 69.4±22.1 days at 0.25 m depth to 84.7±25.2 days at 0.35 m depth. In each year, the spatial pattern of the model simulated zero-curtain period also corresponds well with the snow accumulation pattern indicated from the MODIS SCE observations leading up to full snow cover conditions (Fig. 7).

During the study period, the spatial pattern of model simulated soil freeze onset and zero-curtain period trends at 0.25 m and 0.35 m is closely associated with the trends of MODIS snow onset (Fig. 8); areas with earlier snow onset generally show later freeze onset and longer zero-curtain period. Further analysis indicates that early snow accumulation is the main control on the





soil freezing for the upper (~< 0.4 m) active layer, while zero-curtain period of the deeper soils is more related to ALT (Fig. 9). The model simulations show an overall longer zero-curtain period in areas with deeper ALT, relative to areas with shallower ALT. Approximately 55% of the Arctic Alaska has a maximum thaw depth between 0.45 m and 0.55 m. In those areas, correlations between MODIS snow onset and zero-curtain period generally decrease with increasing soil depths below 0.4 m,

which corresponds with increasing positive correlation between ALT and zero-curtain period. This reduction of the correlation between seasonal snow cover and zero-curtain period is more pronounced in areas with shallower ALT. Compared with the zero-curtain period, we found a much weaker correlation between model simulated soil freeze onset in near-surface soils (< 0.1 m) and observed MODIS snow cover onset, but similar relationships for soil depths below (> 0.2 m) (Fig. S14). This is due to a positive correlation between surface freeze onset derived from MODIS LST and MODIS snow onset (R=0.71, p<0.1)

in Arctic Alaska during the study period; earlier surface freezing leads to cold underlying soil, while earlier snow onset generally leads to warm soil in this area. Therefore, soil freeze onset at near-surface soils has weaker correlations with MODIS snow onset comparing with the middle of active layer (~0.2-0.4 m), while the soil freeze onset subjects to similar controls as the zero-curtain period with soil depth increases. These results indicate that land surface temperature or F/T status may be a relatively poor indicator of soil freezing status and the zero curtain period in the deeper active layer for the Arctic region.

## 4. Discussion

### 4.1 Sensitivity of active layer freezing process to recent climate change

Our results show a strong correlation between the active layer freezing process and snow accumulation in Arctic Alaska during early snow season, especially within the upper (~< 0.4 m) soil layers. Earlier onset and establishment of a complete snow cover generally delay active layer freezing and promote a longer zero-curtain period (Figs. 7&8). Previous studies have highlighted

the decoupling of surface air temperature and soil temperature during the winter season in the northern high latitudes due to the strong insulating effects of seasonal snow cover (Morse et al., 2012; Throop et al., 2012; Koven et al., 2013; Smith et al., 2016). Changes in the rate of accumulation, timing, duration, density and amount of seasonal snow cover play an important role in determining how soil F/T dynamics responds to surface warming (Zhang, 2005; Lawrence and Slater, 2010). However, the relationship between autumn snow onset and soil warming may be variable depending on the timing of snowfall and local

climate conditions (Yi et al., 2015). Early snow onset may enhance thermal buffering of cold surface temperatures, and promote soil warming in colder climate zones such as Arctic Alaska. A shorter snow cover season may cool the soil in colder areas due to less insulation from cold temperatures, but may warm the soil in warmer areas by promoting greater heat transfer into soils (Lawrence and Slater, 2010). The snow cover impact on soil F/T dynamics will also depend on the differences between the timing of first surface freeze and snow cover establishment, especially for the near-surface (~<0.1 m) soils (Kim et al., 2015).

Our model simulations also show that the influence of snow cover on soil freezing is weaker for deeper soil layers (~< 0.5 m) where the freeze onset and zero-curtain period are more closely related to the summer maximum thaw depth (i.e. ALT, Fig. 9





& Fig. S14). This can be largely explained by a close link between ALT and the soil freezing lag rate of the deep soils. During fall, the active layer can freeze back from both the surface and the bottom due to its close contact with the permafrost table (Outcalt et al., 1990; Oechel et al., 1997; Zona et al., 2016). This can be seen from the negative soil freezing lag rate (related to the differences of the soil freeze onset between two adjacent soil layers) at the bottom of the active layer at the GTN-P sites

(Fig. 5 c-d), indicating that the bottom of the active layer freezes first. Increases in the ALT can lead to abrupt changes in the soil freezing lag rate at the same soil depth, which can change from a negative value to a small positive value; this can result in abrupt changes in the zero-curtain period of the bottom active layer, e.g. at the WD1 and WDN sites (Fig. S9b). Previous studies have also reported a delayed soil freeze-back and thus longer zero-curtain with increasing ALT (Morse et al., 2012; Euskirchen et al., 2017). Based on the GTN-P site measurements, deeper soils show a general delay in soil freeze onset relative

to shallower soil layers, with a mean lag rate of 0.79±0.52 days cm$^{-1}$ at depth of 0.35 m; large variability in the soil freezing lag rate is likely associated with different soil structure and variations in active layer soil moisture content (Throop et al., 2012). Therefore, a deepening active layer associated with climate warming will very likely lead to a longer zero curtain period in the deeper soils.

However, the potential response of active layer freezing process to changing climate is likely more complex in the Arctic. The Arctic is expected to experience continued warming and precipitation increases under projected climate trends (Solomon et al., 2007). Both surface warming and a changing precipitation regime can modify seasonal snow cover conditions, leading to a non-linear response of soil temperatures to warming (Lawrence and Slater, 2010; Yi et al., 2015). Increases in winter precipitation and deepening of the snowpack may enhance soil warming, while a reduced snowpack, due to precipitation

decreases or warming-enhanced snow cover loss, may promote soil cooling. More frequent and intense rain-on-snow events during fall and early winter have been observed across the Arctic region with recent warming trends (Ye et al., 2008; Langlois et al., 2017). Therefore, how these climate trends will affect soil moisture and thermal dynamics is a key challenge for accurately estimating soil F/T dynamics and potential carbon and climate feedbacks. In addition, with continued warming and deepening of the active layer, the bottom of the active layer may not freeze-back in the later winter, resulting in a perennially

thawed subsurface soil layer or talik zone; once talik forms, it can greatly accelerate permafrost degradation and result in large changes in surface hydrology and soil carbon decomposition (Yoshikawa et al., 2003; Parazoo et al., 2018).

Large uncertainties may be associated with our model simulations, particularly the soil moisture effects on soil heat transfer during the soil F/T period. Changes in liquid water content during soil freezing varies for different soil conditions, while

accurate simulation of this process is challenging due to comlicated processes controlling ice formation, liquid water movement and heat transfer in frozen soils (Outcalt et al., 1990; Romanovsky and Osterk5mp, 2000; Schaefer and Jafarov, 2016). Our study used in-situ soil dielectric constant ($\varepsilon$) measurements to parameterize the unfrozen water curve and determine the temperature threshold used to define soil freeze onset and calculate the zero-curtain period. However, our results also show large $\varepsilon$ variability in response to freezing temperatures at the SoilSCAPE nodes; the relationship between $\varepsilon$ and the liquid water





content in organic-rich soils may also be substantially different from mineral soils (Engstrom et al., 2005; Mironov et al., 2010), which may not be adequately represented by a static relationship for different soils. On the other hand, potential soil moisture redistribution with active layer deepening is not accounted for in the current model, though this effect is likely small during the study period due to small ALT trends indicated by both the model simulations (Yi et al., 2018) and in-situ observations. In addition, increasing disturbance from thermokarst and wildfire in the ABZ will alter microclimate and soil moisture conditions, vegetation cover and SOC stocks, which are all closely related to the dynamics of ground-ice evolution and permafrost degradation (Grosse et al., 2011; Liljedahl et al., 2016), but not addressed in this study.

## 4.2 Potential of using remote sensing measurements to improve regional monitoring of soil F/T process

Large-scale satellite observations and global reanalysis data generally have difficulty capturing finer-scale snow cover variations and associated impacts on soil F/T dynamics and active layer thermal regime. These limitations are exacerbated in the Arctic due to paucity of regional climate stations and spatially complex microclimate and snow cover properties influenced by local topography, vegetation and winds (Liston and Sturm, 2002; Gisnas et al., 2016). Optical satellite remote sensing, including Landsat and MODIS sensors, can provide accurate local scale information on snow cover extent, though effective regional monitoring from these observations is constrained by persistent cloud cover, atmospheric aerosol contamination and reduced solar illumination for much of the year; moreover, these observations do not include snow depth or water content information, which are critical parameters for hydrologic and ecological applications (Brown et al., 2010; Painter et al., 2016). Snow-covered areas attenuate the emitted microwave radiation from the underlying surface, while the magnitude of microwave emissions and attenuation depends on sensor frequency, snow liquid water content, snow grain size and shape; thus snow water equivalent may be derived from passive microwave sensors, albeit at relatively coarse spatial scale. However, its accuracy is limited in deep snow pack conditions, and its applicability is limited in forest areas and wet snow conditions (Frei et al., 2012). Compared with passive microwave sensors, active radars or scatterometers are capable of much higher spatial resolution and can be particularly useful for regional snow mapping. However, the use of radar is limited to wet snow conditions due to its great penetration in dry snow conditions and strong sensitivity to liquid water content (Dietz et al., 2012). Airborne laser altimeters (lidar) show great potential in mapping snow depth at very fine resolution (Deems et al., 2013; Painter et al., 2016); however, no satellite lidar is currently available for snow mapping. In the near term, potential significant improvements likely will come from merging in-situ and modelling datasets with multi-sensor snow products (Takala et al., 2011; Painter et al., 2016). As an example, this study develops a method using MODIS SCE data for spatial downscaling of coarser global reanalysis snow depth data; however, most improvements were expected to occur during the transitional season with partial snow cover (Fig. 3).

Another major challenge for regional permafrost modelling is the lack of information on subsurface properties, particularly for organic soils with distinct hydraulic and thermal properties, which are prevalent in the ABZ. Current permafrost models generally use regional or global SOC inventory data to parameterize the SOC profiles following an exponentially decreasing





curve (Lawrence and Slater, 2008; Rawlins et al., 2013; Yi et al, 2018). However, large discrepancies are apparent from the available SOC inventory records in the ABZ (Liu et al., 2013; Hugelius et al., 2014). There is also large regional variability in the vertical SOC distribution due to multiple processes affecting the SOC distribution in cryoturbated soils (Mishra et al., 2013; Hugelius et al., 2014; Hossain et al., 2015). Radar backscattering measurements are directly sensitive to soil dielectric

properties, which are strongly correlated with soil moisture, texture, and F/T data (Dobson et al., 1985; Mironov et al., 2010; Bartsch et al., 2016). Our model experiments and analysis using in-situ dielectric constant measurements and radar retrieved soil dielectric constant to characterize soil freezing process, albeit simple, show the potential of longwave radar remote sensing in mapping of soil carbon content, active layer F/T and moisture profiles. This may enable improved model representation of subsurface processes affecting soil F/T processes. However, similar to all other inversion problems, radar retrievals suffer from

ambiguity in the inversion parameter definitions mainly due to insufficient information about the subsurface profile (e.g. Tabatabaeenejad et al., 2015; Chen et al., in review). Therefore, new methodology is needed to address the underdetermined nature of the radar backscatter inversion and associated land parameter retrievals, by either including additional observations or other synergistic information from soil physical models to reduce parameter dimensions in the radar model (e.g. Sadeghi et al., 2016). The vegetation canopy also has a large impact on the radar backscatter, while separating the radar contribution of

subsurface soils from the vegetation canopy remains a challenge. A modelling framework that can ingest multi-sensor remote sensing data such as vegetation parameters retrieved from lidar and a radar backscatter forward model may provide a promising approach to address the above limitations. Additional airborne radar sampling targeting regional disturbance gradients may also provide the necessary information for the regional modelling framework to represent increasing disturbance regimes and associated impacts on active layer F/T dynamics in the ABZ.

**5. Conclusions**

In this study, we used a remote sensing driven permafrost model and a newly developed high-resolution snow dataset to simulate the active layer freezing process, including soil freeze onset and the zero-curtain period in Arctic Alaska during the recent satellite period (2001-present). The model simulations were combined with multiple in-situ measurements, and local-scale soil dielectric constant retrievals derived from airborne longwave (P-band) radar data to investigate the regional

sensitivity of the soil freezing and zero curtain period to recent climate change. Our results indicate that: 1) the soil freeze onset and zero-curtain period in the upper soils (< 0.4 m) are primarily affected by early season snow cover accumulation, whereby areas with earlier snow onset generally show delayed soil freeze onset and prolonged zero-curtain period; 2) the influence of early season snow cover on the soil freezing gradually decreases with increasing soil depth and the zero-curtain period of deeper soils (> 0.5 m) are more closely related to the thickness of active layer due to increasing delay in soil freezing

with active layer deepening. Therefore, deepening active layer associated with climate warming will very likely lead to longer unfrozen period in the deeper soils and potentially result in more carbon loss during the cold season. These findings highlight the importance of relatively fine-scale snow cover and active layer thickness products for better understanding of potential



carbon and climate feedbacks in permafrost ecosystems. Our model experiments and analysis using in-situ and radar retrieved dielectric constant data to characterize soil freezing process, albeit simple, show the potential of longwave radar remote sensing in mapping of soil carbon content, soil active layer F/T and moisture profiles. This may enable substantial improvements in the way models represent key subsurface processes at permafrost landscapes, enabling more accurate predictions of boreal and arctic environmental changes.

*Data availability*. The regional model simulations will be archived and distributed for public access through the NASA ABoVE archive at the NASA ORNL DAAC (https://daac.ornl.gov/). The radar retrievals are available upon request. Other data used in this study were obtained from free and open data repositories.

*Author contributions*. Y.Y. and C.E.M. initiated the study. Y.Y did the calculations and wrote the paper. R.H.C. and M.M. contributed to the data. All authors contributed to the discussions and provided feedbacks on the final version.

*Competing interests*. The authors declare no conflict of interest.

*Ackowledgements*. Funding for this study was provided by NASA (NNX15AT74A). The authors thank V. Romanovsky's group for providing the GTN-P soil temperature data in Alaska. A portion of this research was carried out at the Jet Propulsion Laboratory, California Institute of Technology, under contract with NASA. © 2018. All rights reserved.

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



**Table 1: The list of key parameters used in this study.**

| Parameters | Description | Range | This study | Note |
|---|---|---|---|---|
| b | Shape parameter for the unfrozen water content curve (Eq. 2) | 0.1 ~ 1.0 | 0.63 | Romanovsky and Osterkamp (2000); Schaefer and Jafarov (2016) |
| $P_{cutoff}$ | Critical value for snow probability occurrence (Eq.6) | 0.4 ~ 1.0 | 0.5 | Determined using trial and error method |
| δ | Critical value of soil dielectric constant changes at freeze-up (Eq. 9) | 0 ~ 50% | 15% | Determined using trial and error method |
| $T_{cutoff}$ | Critical value of soil temperature at freeze-up | 0 ~ -1.0°C | -0.35°C | Determined using SoilSCAPE measurements |



**Table 2: The accuracy of the spatial filter algorithm applied to Aqua MODIS SCE product during spring (from April to June) and fall (from September to November) transitional season in Alaska averaged from 2003 to 2015. The pixels that were cloud contaminated in Aqua MODIS, but indicated as clear conditions in Terra MODIS were used for evaluation. The percentage of cloud contaminated and evaluating pixels were calculated for the entire Alaska, while the accuracy and misclassification were calculated as the percentage of the evaluating pixels. The results based on Terra Aqua MODIS SCE product show similar accuracy, while only the results based on Aqua MODIS were shown here due to a higher percentage of cloud contaminated pixels (available for evaluation) in Aqua images.**

| | Spring transitional season | | | Fall transitional season | | |
| | April | May | June | September | October | November |
|---|---|---|---|---|---|---|
| Cloud contaminated pixels (%) | 5.0±2.7 | 11.2±2.6 | 4.4±2.3 | 8.8±4.0 | 13.2±3.7 | 11.3±3.5 |
| Evaluating pixel (%) | 3.5±1.7 | 5.8±1.1 | 2.4±1.1 | 5.0±2.0 | 9.1±2.3 | 7.2±2.1 |
| Accuracy (%) | 92.0±3.3 | 80.0±1.9 | 82.9±4.0 | 81.8±2.3 | 86.2±3.5 | 97.0±2.2 |
| Misclassification of land pixels (%) | 4.1±1.4 | 8.2±0.9 | 6.7±1.4 | 8.6±1.1 | 6.7±1.4 | 2.0±1.3 |
| Misclassification of snow pixels (%) | 4.0±2.0 | 11.8±1.3 | 10.3±2.9 | 9.6±1.9 | 7.1±2.2 | 1.1±1.0 |





**Table 3: Statistics of 1-km MERRA-2 snow depth data generated using different spatial interpolation schemes compared with in-situ observations at Alaskan SNOTEL sites.**

| | num of sites | R | | Bias (m) | | RMSE (m) | |
|---|---|---|---|---|---|---|---|
| | | Yi et al (2018) | this study | Yi et al (2018) | this study | Yi et al (2018) | this study |
| Arctic Alaska | 8 | 0.84 | 0.85 | -0.03 | -0.01 | 0.18 | 0.16 |
| Other areas <400 m | 19 | 0.78 | 0.81 | 0.01 | 0.01 | 0.39 | 0.28 |
| 400-800 m | 18 | 0.86 | 0.91 | -0.01 | -0.09 | 0.44 | 0.41 |
| >800 m | 10 | 0.84 | 0.88 | -0.08 | 0.02 | 0.32 | 0.27 |



**Table 4: Comparisons of model simulated and in-situ observed zero-curtain period at depth of 0.35 m for sites along the DHN transect. The model simulated mean zero-curtain period was calculated from 2001 to 2016. Zero-curtain period calculated using in-situ soil temperatures at adjacent sites (e.g. the three Franklin Bluff sites, Table S1) were generally very close, and thus were combined for a longer observational record.**

|  | WD | DH | FB | SagMAT | SagMNT | HV | GL |
|---|---|---|---|---|---|---|---|
| num of year | 10 | 8 | 15 | 8 | 9 | 7 | 11 |
| R | 0.62* | 0.67* | 0.82* | 0.87* | 0.71* | 0.48 | 0.73* |
| Bias (days) | 14.40 | 6.25 | -5.27 | 14.12 | 18.11 | -13.85 | -1.83 |
| RMSE (days) | 19.15 | 19.38 | 13.73 | 15.83 | 26.5 | 22.20 | 18.75 |
| Model mean zero-curtain period (days) | 64.5 | 77.0 | 63.5 | 71.0 | 79.5 | 71.0 | 83.0 |

5   Note:  *denotes p<0.1; WD: West Dock; DH: Deadhorse; FB: Franklin Bluffs, SagMAT: Sagwon MAT, SagMNT: Sagwon MNT; HV: Happy Valley; GL: Galbraith Lake. Zero-curtain period at Imnaviat 1 and GL sites were very close, and thus combined to form a longer time series.



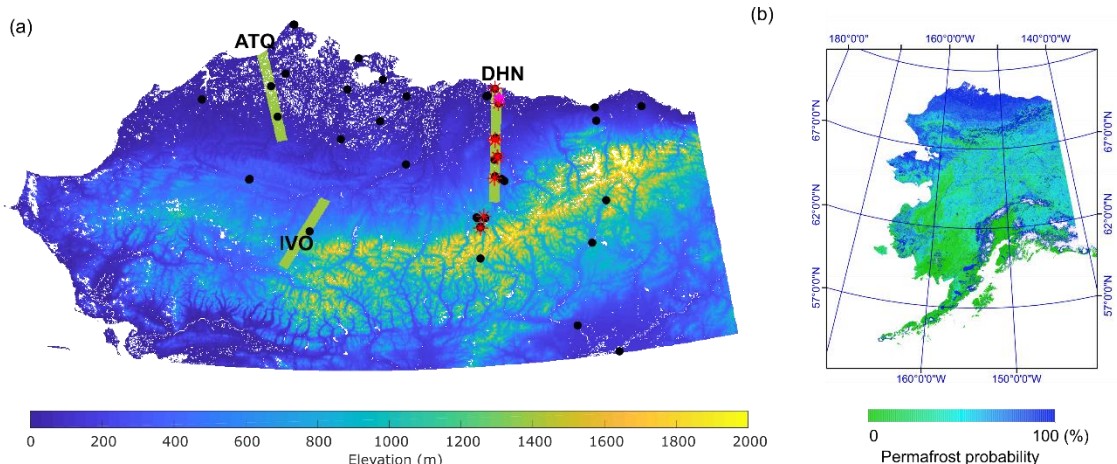

**Figure 1: a) The study area (Arctic Alaska, >66.55° N) and locations of different datasets; b) the permafrost distribution in the Alaska state based on a satellite and soil inventory based permafrost probability map (Pastick et al., 2015), indicating higher permafrost occurrence in Arctic Alaska than other areas of the state. The in-situ sites include the Prudhoe Meadow SoilSCAPE site (magenta star), GTN-P soil temperature sites (red starts) and CALM ALT sites (black dots). AirMOSS P-band radar data were obtained in late August and early October in 2014 and 2015 and shown as yellowgreen lines. There is also a radar flight in Barrow area, but not included for analysis due to a large fraction of surface water bodies there.**





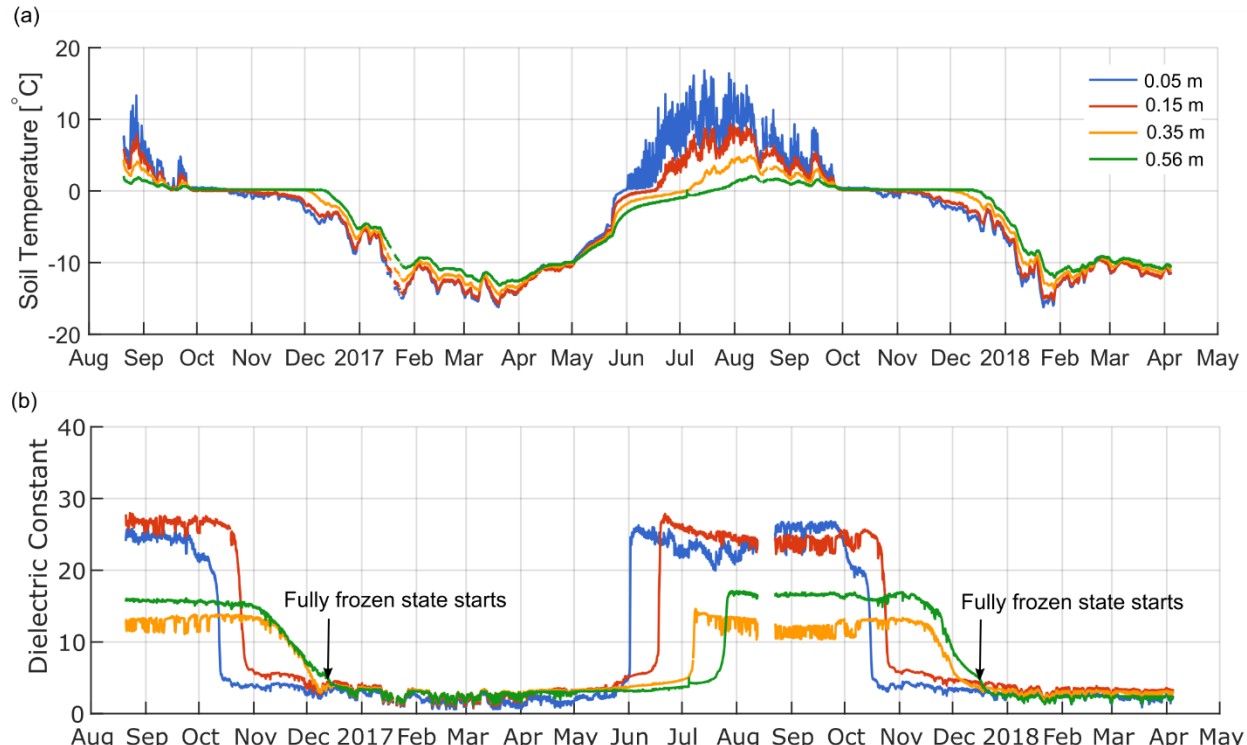

**Figure 2: The in-situ measurements of soil temperature (a) and dielectric constant (b) at one of the sensor node (S6) of the Prudhoe Meadow SoilSCAPE site (http://soilscape.usc.edu).**





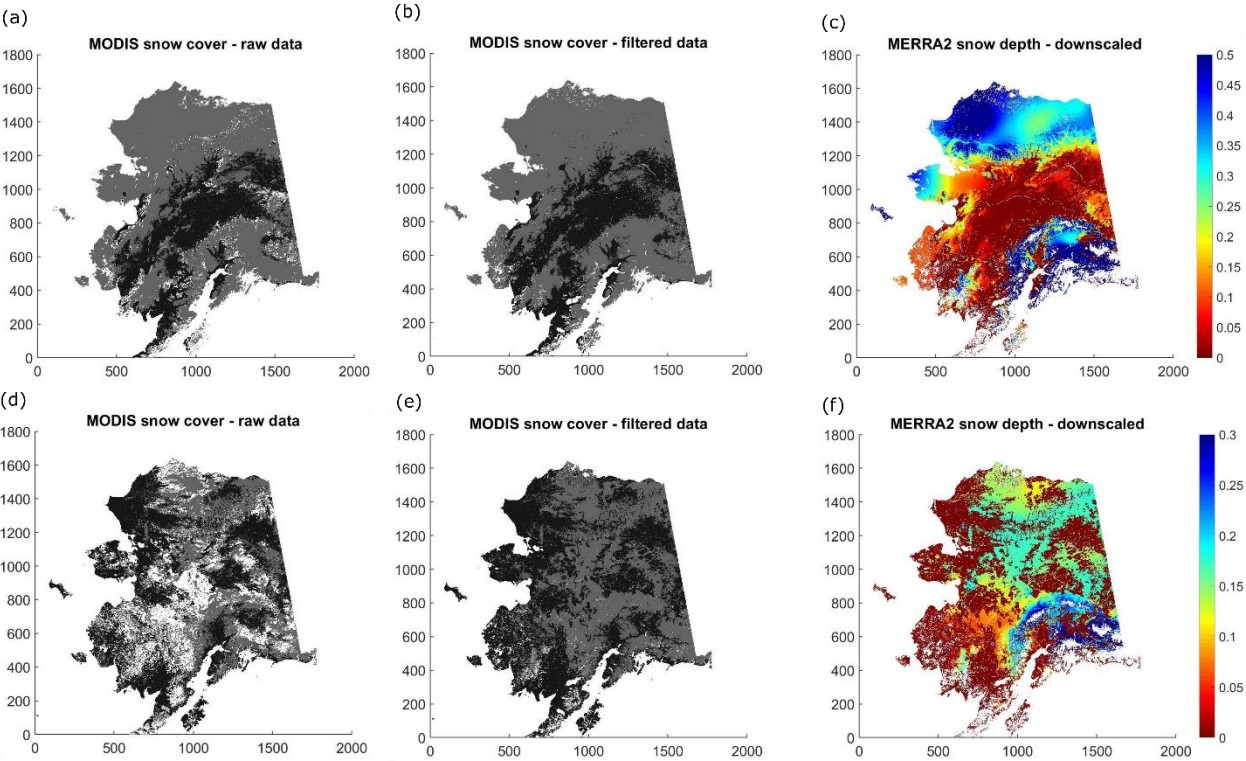

**Figure 3: Illustration of the snow data processing: the raw (a&d) and cloud filtered (b&e) MODIS SCE images using an elevation-based spatial filter and downscaled MERRA-2 snow depth data (c&f) using the filtered MODIS SCE and DEM data during snow melting (top: 04/23-04/30) and early snow accumulation period (bottom: 09/30-10/07) in 2007. In the MODIS images, snow covered areas were shown as gray, while land and cloud covered areas were shown as black and white respectively.**



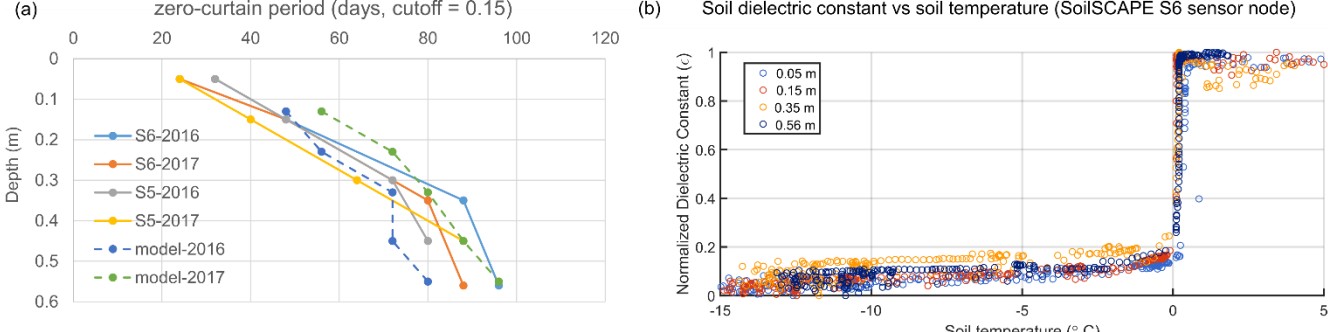

**Figure 4: Soil freezing characteristics at the Prudhoe Meadow SoilSCAPE site: (a) comparisons of model and observed zero-curtain period at two sensor nodes (S5 and S6) using a cutoff threshold of 0.15 (Eq. 9) for both in-situ soil dielectric constant and model simulated unfrozen water content to determine soil freeze onset; b) the changes in in-situ soil dielectric constant during soil freezing process at the S6 node. The soil dielectric constant was normalized using the maximum and minimum value of ε throughout the observation period.**



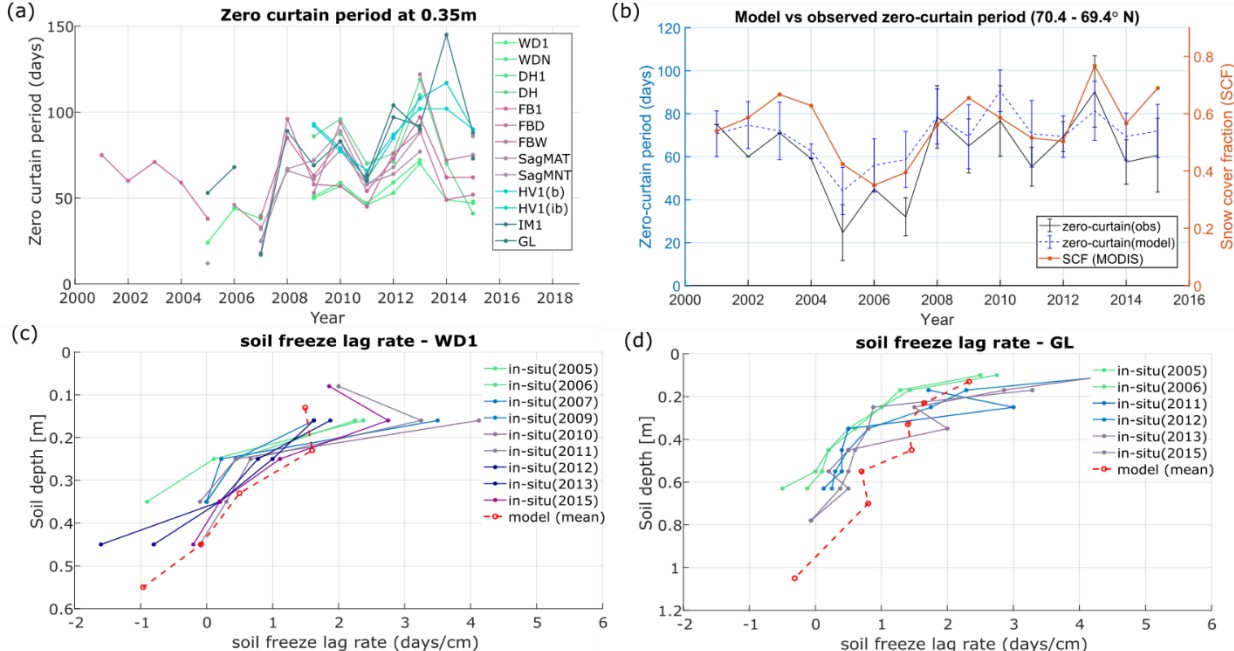

**Figure 5: Comparisons of soil freezing characteristics derived from GTN-P soil temperature measurements and model simulations along the DHN transect: a) the inter-annual variations of zero-curtain period derived from in-situ measurements; b) variations of model versus in-situ zero-curtain period with MODIS SCF averaged for the northern part of the DHN transect; c-d) changes in the soil freeze lag rate with depth at two sites. Both sites have lower ALT in the earlier years shown here, with 0.37 m during earlier period versus 0.43 m during later period for WD1 site, and 0.47 m versus 0.57 m for GL site.**





**Figure 6: Soil freeze process indicated by the radar retrieved (Chen et al., 2017; in review) soil dielectric constant (ε1) of surface soils (~<0.10 m) at the DHN transect in August (a) and October (b), and changes in ε1 in relation to MODIS SCF (c). The ε1 differences between August and October were binned to 0.1° latitudinal bin, while SCF was calculated as the percentage of snow covered pixels indicated by MODIS SCE data for each 0.1° bin. The standard deviation of ε1 differences for each 0.1° bin was shown as error bars.**

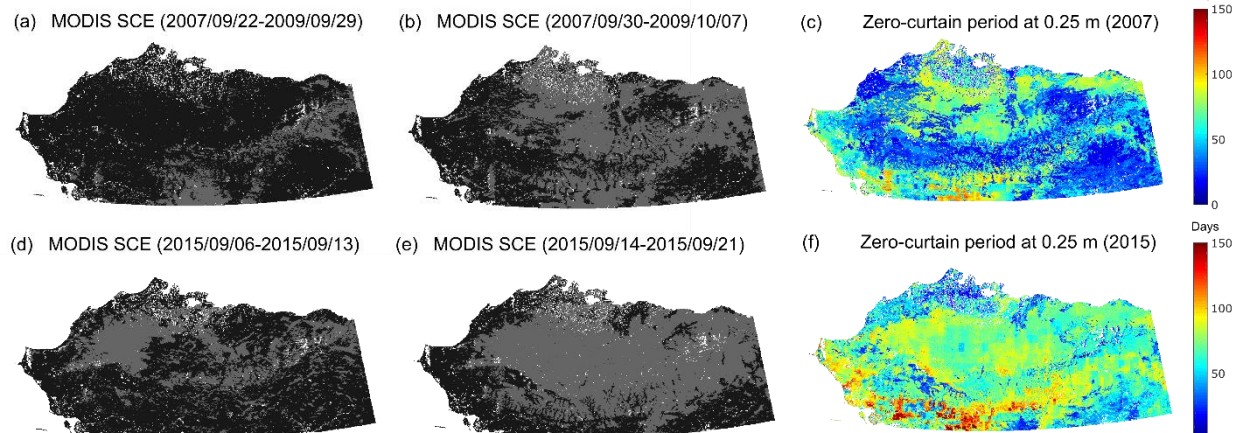

**Figure 7: Model simulated zero-curtain period at 0.25m in relation to snow accumulation during early snow season indicated by filtered MODIS SCE images in two years with later (2007: a-c) and earlier (2015: d-f) snowfall. In the MODIS images, snow was shown as dark gray, while land was shown was black and the areas masked out were shown as white.**





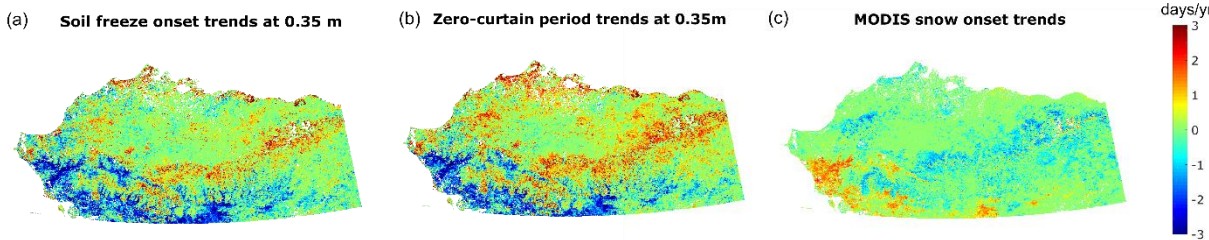

**Figure 8: Trends of model simulated soil freeze onset (a), and zero-curtain period (b) at depth of 0.35 m and MODIS snow onset date (c).**





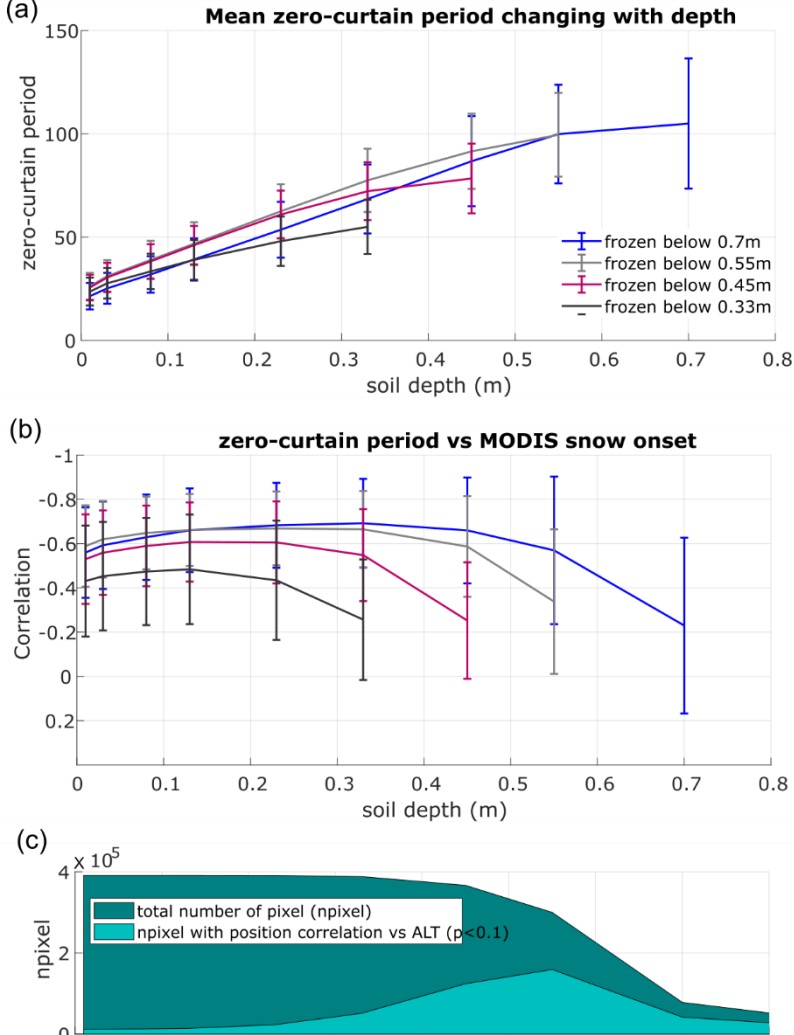

**Figure 9: Regional statistics of model simulated zero-curtain period (a) and its sensitivity to MODIS snow onset and model simulated ALT (b-c) from 2001 to 2016: a) regional mean (2001-2016) of model simulated zero-curtain period at different depths; b) changes in correlations between snow onset and zero-curtain period with depths; for both a) and b), the study area was divided into 4 groups: soil column froze below 0.33 m, 0.45 m, 0.55 m, and 0.7 m. The soil column of the majority of the study area froze below 0.7m. c) the proportion of pixels with significant positive correlation between zero-curtain period and ALT at different depths. The total number of unfrozen pixel was shown as "npixel".**