# Peer review of "Sensitivity of active layer freezing process to snow cover in Arctic Alaska"

_The Cryosphere, 2018_

## Referee Comment (RC1) · Anonymous Referee #1 · 3 Oct 2018

The manuscript presents a study on soil active layer dynamic sensitivity to snow cover from an improved remotes sensing driven permafrost model, which used several remote sensing information, including MODIS snow cover data. The modeled snow depth, soil freeze-up and zero-curtain period were calibrated and validated against in situ observations. The model was used to evaluate the sensitivity of active layer freezing to snow cover conditions.

The manuscript present interesting and valuable results on the relation between snow cover and active layer evolution. However, the manuscript is sometime difficult to read and some methodological section could be improve. The many supplement figure make the manuscript even harder to follow and I would suggest to lightening it. I recommend the publication of the manuscript following revisions:

[Figure]

1. p1 - line 24 : "this was also consistent with findings based on the airborne radar ÉŻ retrievals in 2014 and 2015" not clear in the context.

2. The introduction is very well written and interesting. However, at the last paragraph, it is not clear what the objectives of the paper is. Because there are many results and analysis in the manuscript, a clear definition of the objectives would help the reader.

3. p4 – l 3-4 : does the model as a name? (minor: sentence copy pasted from last paragraph)

4. p5 – Line 13 : At that point, it is not clear how SCE data can improve snow depth data? SCE is used to remove the snow where the model still simulate snow cover? It should be clarify at that point how you come to correct snow depth from binary information.

5. p8 – l13: At what frequency is ÉŻ. In addition, it would be important to clearly distinguish the ÉŻ (SoilSCAPE) and ÉŻ1 (radar), and their frequency. Maybe use more precise acronym?

6. p9 – l6: How the model thermal properties were adjusted (changing the thermal conductivity of the soil or changing the amount of organic matter in the soil?)

7. 2.3.3 Airborne radar retrievals: The title is not clear. Maybe "Airborne radar permittivity retrievals"? The section is not very clear as well. 1) At this point it is not clear how the dielectric will help to analyse the relation between active layer freezing and snow cover. 2) What is the expected penetration depth with P-band (does it change with frozen vs unfrozen)? 3) L17 : define airborne land parameter. 4) "an iterative model inversion" .: what model (radar scattering model, but which one?). 5) p10 – l1 : does soil moisture variation can have an impact?

8. At the end, the usefulness of P-Band radar is not clear. The results obtain with P-Band radar are limited (Figure 6) but add to the complexity and the length of the manuscript.

[Figure]

9. 3.1 Model Validation: "model parameterization and validation" would be more appropriate.

10. p12-28 : Have you done an inversion to get that number? Clarify how you calculated the 15%.

11. p15 – l13: Why higher elevation could impact the relationship? This point should be develop more. Is the elevation impact ÉŻ1 or MODIS SCF, or the soil thermal regime? At what point it is in opposition to the conclusion.

12. Fig S13: I do not see how Fig. S13 shows the effect of snow accumulation. Need more explanation.

13. 3.2.2 first paragraph: why considering only two years for this analysis? Does using many years would show that the mean snow onset is correlated to mean zero-curtain period? The results would be much stronger?

14. The model evaluation show some uncertainties in the model. At what point these uncertainties could influence the results and the conclusion?

Minor :

p19 – l5 : "information" instead of data?

Figure 1: The magenta and the red stars are very similar.

Figure 5 : (b) at 0.35 m?

---

## Referee Comment (RC2) · Anonymous Referee #2 · 3 Dec 2018

General:

The paper by Yi et al. presents a study applying a remote sensing driven permafrost model, in situ observations and airborne P-band SAR retrievals to study the influence of snow cover characteristics on permafrost active layer dynamics in Alaska. The model, developed by the same group, is driven with diverse observational data including MODIS LST and SMAP L4 root soil moisture products. A part of the paper is devoted to downscaling MERRA-2 snow depth estimates to 1 km resolution using the MODIS SCE record and a digital elevation model, and assessing the validity of the downscaled product to in situ data. Along a limited study area, the authors report the autumn zero-curtain period predicted by the model to closely match observations, while both of these were found to be positively correlated with fractional snow cover.

[Figure]

Some supporting evidence is derived from the P-band SAR observations, although the analysis here is only perfunctory. In regional scale simulations, it is reported that the top active layer zero-curtain period is correlated with timing on snow cover onset, while deeper layers are more influenced by maximum thaw depth. The authors indicate that with climate induced deepening of the active layer this may result in a feedback of an extended unfrozen period in the lower active layer, resulting in increased carbon loss during winter. The theme of the paper is of high interest to the cryosphere community, as bridging the gap between what can be observed via remote sensing and what actually happens with permafrost is a long standing but still ongoing topic. Different proxies such as snow cover and surface freeze thaw driving physical models are required due to the difficulties of directly observing deep soil processes. This study presents a valuable step forward in modeling permafrost dynamics with aid of remotely sensed observations, which is why I feel the study merits to be published in the Cryosphere. Downscaling methods applied to MERRA-2 snow depth estimates may be of interest when applied to other coarse-scale products on snow depth and mass as well. The paper is well written, although parts of it are rather heavy reading (mainly section 2) and could benefit from cutting down some of the text, replaced by perhaps illustrations depicting the analysis process in the form of flowcharts etc. The authors may also consider the usefulness of some of the results, in particular the added value of the P-band SAR observations. Are these really needed, or do they provide only an unnecessary diversion? In any case I recommend publication of the work, following response to the following minor comments.

Minor comments:

1. Introduction, P3, lines 6-8: I think citing "high-frequency" instruments here is a bit misleading, as it refers mainly to the FT-ESDR dataset (where high-frequency Ka band data is indeed applied). This statement is not really applicable to scatterometry (e.g. C-band ASCAT) or L-band SMOS and SMAP F/T products. It is a good question how much "high frequencies" indeed tell about soil freezing due to very limited penetration

already at C-band in surface vegetation, and the influence of e.g. snow cover at frequencies above Ku band. Of course, the field of view of all of these instruments is very large resulting in coarse-scale products, so that part of the sentence applies (but what about C-band SAR, which are increasingly available from Sentinels and RCM?) I suggest to rewrite this sentence properly acknowledging the various benefits/caveats of different wavelengths. Also add a suitable reference to the scatterometry-based method as well as the SMOS soil F/T product.

2. Section 2 requires some effort from the reader. It is difficult to follow all the diverse steps required for first gap filling and downscaling MODIS and MERRA-2 snow products, steps required for other reference data, followed by the actual data analysis, and where all these data are finally applied. I understand this is due to the complexity of the analysis, but still it took me two to three read-throughs to finally get a grasp. A flow chart summarizing all of these data preparation steps and the consecutive analysis, indicating clearly where each bit of data is used, could greatly clarify the process. This could be supported by a brief introduction on what is to follow in the beginning of section 2 (following for example the one you have in the beginning of section 3).

3. P4 eq. 1. For clarity please define all variables in the equation, even though these may be self-evident ($z$, $t$, $z_s$, $z_b$ etc.). Furthermore it is not clear how volumetric heat capacity and thermal conductivity are calculated (it is only stated that these "vary" with F/T state, depth etc). Please clarify in the text.

4. P7 eq. 7. Again, please give all variables of the equation in the text.

5. P10 line 20. "Mostly focusing" is a bit unclear… can you elaborate? Where are other transects besides DHN used, could be stated here.

6. P11 line 21 "a cutoff threshold of 50%" is not introduced or explained anywhere from what I can see. A reader not familiar with the concept might be highly confused. Please elaborate.

7. Section 3.1.2, p12 lines 18-26. Might parts of this paragraph be more suited in the methods section? Also, please explain further how thermal conductivity was assumed to "gradually increase". Was the increase linear? This might also give a partial answer to my comment 3, and the explanation might be also better suited for section 2.

8. P13 line 15: "in situ dielectric measurements in frozen soils have significant uncertainties". Why? This should be explained, and references given to support the claim. Also, avoid use of word "significant" unless you are discussing statistical significance.

9. Section 3.2.1. is the inclusion of the P-band airborne SAR observations really necessary? Although the results seem to corroborate your findings, I think these data might be of more interest as a separate paper by itself (provided more analysis can be provided; the data would seem useful for e.g. electromagnetic model exercises from frozen soil). The title of the section is at least strange, change that if nothing else.

10. P18 line 19 add some suitable references to passive microwave products already here (also e.g. works by Kelly et al.).

11. P18 line 22 "the use of radar is limited..." well this limitation applies to current operational systems as they typically have X-band as the highest frequency. However, QuikSCAT provided already some indication that snow volumetric properties could be captured using radar. Airborne data (e.g. Yueh et al., 2009; King et al., 2018) have provided similar indications (while also revealing limitations). What I am driving at: please be specific about the wavelength and that you are talking about radars currently in space when you make this claim.

12. P18 line 25 "no satellite lidar is currently available..." IceSAT2 just went up, and although no terrestrial snow products are immediately planned, it could be of use also for snow mapping (despite limited coverage). Any comment? Might be good to acknowledge the mission.

13. P19: "Radar backscattering measurements..." again I miss definition of the wavelength. A suitably long wavelength would be needed to get meaningful information from subsurface properties, while shorter wavelengths would provide limited information.

14. P19: line 9 "similar to all other inversion problems..." Replace 'all' with 'many' or 'typical'? Can't one can find unambiguous inversion problems?

15. Conclusions, p20. At the end you could cite upcoming or planned long-wavelength radar missions, which may be of use for the purpose you cite (NISAR, TanDEM-L, BIOMASS).

16. Figure 8: add time period for which trends were calculated to figure caption.

Editorial: 1. P14 line 8 "we then discussed" might sound better as "we then discuss" as the discussion is to follow. 2. Figure 9a: add unit (days) to y-axis

---

## Author Comment (AC1) · 31 Dec 2018

**Response to referee's comments on "Sensitivity of active layer freezing process to snow cover in Arctic Alaska"**

**Authors:** Y. Yi, J. S. Kimball, R. Chen, M. Moghaddam, C. E. Miller

*Dear Editor,*

*We appreciate the helpful comments from the two reviewers. We made substantial revisions on the methods section in order to make it easier to follow. We also made an effort to justify why we included the radar data in our analysis. In our future work, we will develop various modeling experiments (combining the soil physical and electromagnetic models) to explore the value of radar (particularly L- and P-band) and radiometer data in frozen soil studies. As an initial effort towards this goal, we included a preliminary analysis of the P-band radar retrievals in this study.*

*Our responses to the comments are provided in the following text, and the revised manuscript is enclosed as a supplement with changes highlighted. Thank you for considering our manuscript.*

**Review 1#:**

*1) General comments: The manuscript present interesting and valuable results on the relation between snow cover and active layer evolution. However, the manuscript is sometime difficult to read and some methodological section could be improve. The many supplement figure make the manuscript even harder to follow and I would suggest to lightening it.*

**Response:**

We made the following revisions to address the reviewer's concern:

-The method section was rewritten and a flow diagram (Fig. 1) was added to make it easier to follow.

-We cut down the number of supplement figures from 14 to 10.

Please refer to the manuscript for details.

*2) p1 - line 24: "this was also consistent with findings based on the airborne radar $\varepsilon$ retrievals in 2014 and 2015" not clear in the context.*

**Response:**

We replaced this sentence with the following sentence to avoid confusion:

Page 1, line 23-25: "We also examined the airborne P-band radar retrieved ε profile along this transect in 2014 and 2015, which is sensitive to near-surface soil liquid water content and freeze/thaw status."

*3) The introduction is very well written and interesting. However, at the last paragraph, it is not clear what the objectives of the paper is. Because there are many results and analysis in the manuscript, a clear definition of the objectives would help the reader.*

**Response:**

We revised the last paragraph of the introduction to clearly state our objectives:

Page 3, Line 23-33: "The objective of this study was to clarify primary environmental controls on the timing of seasonal freezing of the active layer and the duration of the zero-curtain period in Arctic Alaska. A remote sensing driven soil process model was used to examine the impact of climate variability and snow cover properties on the estimated soil F/T transition and zero-curtain within the active layer profile. Model simulations were conducted at 1-km resolution and over a multi-year period (2001-2016) to capture landscape level heterogeneity in active layer freezing process and its sensitivity to regional environmental trends. To better capture the snow cover variability and its impact on soil F/T dynamics, we also developed a new algorithm to generate a fine-resolution (1km) snow depth dataset as soil model inputs through combining the MODerate resolution Imaging Spectroradiometer (MODIS) snow cover extent (SCE) and coarse-resolution global reanalysis data. The timing and duration of frozen soil conditions in the Arctic strongly influence underlying permafrost stability and potential vulnerability of vast SOC stocks in the tundra area (Parazoo et al., 2018; Yi et al., 2018; Zona et al., 2016). Thus, the model

results also help clarify the potential response of cold-season soil respiration and boreal-Arctic carbon cycle to current climate warming trends."

*4) p4 – l 3-4: does the model as a name?*

**Response:**

The model originated from a permafrost hydrology model (Rawlins et al., 2013; Yi et al., 2015), but was revised to have a more flexible structure to use remote sensing data as model inputs or parameterization (Yi et al., 2018). Different from the permafrost hydrology model, the model used in the current study does not simulate soil water transfer, though it does account for soil water phase change. We are further refining the model logic and parameterization to be coupled with a radar scattering model. We will name the model thereafter.

*5) p5 – Line 13: At that point, it is not clear how SCE data can improve snow depth data? SCE is used to remove the snow where the model still simulate snow cover? It should be clarify at that point how you come to correct snow depth from binary information.*

**Response:**

The reviewer may refer to the snow data processing in our previous study (Yi et al., 2018), which is described in the following sentence: "We first interpolated the MERRA-2 data over a finer 1-km spatial grid using an inverse-distance weighting scheme, and then used the MODIS 500 m SCE data to identify snow-free pixels within each 0.5° MERRA-2 grid and adjust the 1-km snow depth estimates accordingly."

In the current study, the cloud-filtered MODIS 1-km SCE imagery was used to describe the snow distribution within each MERRA-2 0.5 grid cell; the snow depth for each snow-covered 1-km pixel was then estimated based on the neighboring MERRA-2 grid cells with weights predicted using an elevation-based spatial filter. To avoid confusion, we added a flowchart (Fig. 1) to describe the snow data processing and added additional clarification on this in Section 2.1:

Page 4-5: "This information can be derived from the MODIS SCE data; however, persistent cloud cover and patchy snow conditions constrain the ability of the MODIS SCE data to capture snow cover variability, especially during the transitional season. To overcome this constraint, in this study we developed an elevation-based spatial filtering algorithm to predict snow occurrence for MODIS cloud contaminated pixels; we then used the cloud-free MODIS SCE data to downscale the MERRA-2 snow depth data (Fig. 1). For each snow-covered 1km pixel indicated by the MODIS data, we estimated the snow depth based on the snow depth of neighbouring MERRA-2 0.5° grid cells, with weights predicted using a similar spatial filter."

*6) p8 – l13: At what frequency is $\mathcal{E}$. In addition, it would be important to clearly distinguish the $\mathcal{E}$ (SoilSCAPE) and $\mathcal{E}1$ (radar), and their frequency. Maybe use more precise acronym?*

**Response:**

We added the following information on the in-situ $\mathcal{E}$ data for better clarification:

Page 9, Line 22: "$\varepsilon$ was measured using a METER TEROS 12 soil moisture sensor operating at 70 MHz."

The NASA UAVSAR airborne P-band radar (also known as AirMOSS) uses a frequency of 430 MHz. Even though the soil dielectric constant (either measured using in-situ sensor or retrieved from P-band radar) very likely has different magnitude due to the frequency differences, its sensitivity to changes in soil liquid water content should be similar within the range of these two frequencies (Mironov and Savin, 2015). Our focus is on the sensitivity of the soil dielectric constant to soil freezing status (changes in liquid water) rather than the magnitude of the soil dielectric constant itself so we did not distinguish between the two terms in the manuscript.

Mironov, V. and Savin, I.: A temperature-dependent multi-relaxation spectroscopic dielectric model for thawed and frozen organic soil at 0.05–15 GHz, Physics and Chemistry of the Earth, Parts A/B/C, 83–84, 57–64, doi:[10.1016/j.pce.2015.02.011](10.1016/j.pce.2015.02.011), 2015.

*7) p9 – l6: How the model thermal properties were adjusted (changing the thermal conductivity of the soil or changing the amount of organic matter in the soil?)*

**Response:**

The model thermal properties were adjusted based on the soil texture (mineral or organic soils) and soil water content (also affected by mineral or organic soils). We added the following text to describe how we adjusted the soil thermal properties:

Page 10, Line 15-19: "The unfrozen soil thermal conductivity within the organic layer was assumed to gradually increase with depth from ~0.5 W m$^{-1}$ K$^{-1}$ in the surface organic soil layer to ~1.2 W m$^{-1}$ K$^{-1}$ at 0.33 m depth for mineral soils, accounting for increases in the soil bulk density (Letts et al., 2000). Similarly, soil porosity was assumed to gradually decease from 0.8 at the surface to ~0.4 in the deeper mineral soil layers. The soil thermal conductivity for frozen conditions can then be determined from Eq. (10)."

Page 8: we also added Eq. (8-10) to provide further details on the model soil thermal parameterization.

*8) 2.3.3 Airborne radar retrievals: The title is not clear. Maybe "Airborne radar permittivity retrievals"? The section is not very clear as well.*
   *(1) At this point it is not clear how the dielectric will help to analyse the relation between active layer freezing and snow cover.*
   *(2) What is the expected penetration depth with P-band (does it change with frozen vs unfrozen)?*
   *(3) L17: define airborne land parameter.*
   *(4) "an iterative model inversion" .: what model (radar scattering model, but which one?).*
   *(5) p10 – l1: does soil moisture variation can have an impact?*

**Response:**

We added the following information on the airborne radar retrievals in Section 2.5 (originally as Section 2.3.3):

- We changed the title to "Soil dielectric constant retrievals from airborne P-band radar"

- We added the following sentences to more clearly define why the airborne radar retrievals add value to our analysis:

Page 11, Line 7-10: "The soil model unfrozen water content curve used to define the soil freeze-up and zero-curtain was only calibrated using limited SoilSCAPE soil dielectric measurements. We therefore evaluated the sensitivity of surface soil dielectric properties derived from local-scale (~50 m) airborne low frequency (P-band) radar acquisitions along regional transects in northern Alaska and associated F/T patterns to snow cover variations."

- We added the following additional information on the penetration depth of P-band:

Page 4, Line 17-18: "…Longwave (P-band) polarimetric SAR (PolSAR) data with larger penetration depth (~50-60 cm depending on soil moisture content) were acquired…". The penetration depth does change with soil F/T status, and increases when the soil freezes, as radar is very sensitive to soil liquid water content."

- We defined the radar retrieved land parameters as:

Page 11, Line 12-13: "Multiple soil parameters, including active layer thickness (ALT) and soil moisture (converted from soil dielectric constant),…"

- The radar scattering model used for the land parameter retrieval is presented in Chen et al. (2018; in press). We clarified this in the manuscript:

Page 11, Line 21-22: "…radar scattering model simulations using the above three-layer soil dielectric model."

- Based on the original text (P10, Line 1: "…frozen conditions of surface soils. Initial validation indicated that the radar retrieved ALT along the Dalton highway transect show…", we are not sure what retrievals (ALT or soil dielectric constant) the reviewer was asking for:

1) If it is for ALT, this was discussed in two other related papers (Chen et al., in press; and Yi et al., 2018), where the sensitivity of P-band radar to ALT is reduced when the active layer is saturated.

2) If it is for radar retrieved soil dielectric constant, soil moisture (liquid fraction) directly affects the value of $\varepsilon$. However, in this study, we examined the relative changes in $\varepsilon$ between August and October, which is most sensitive to changes in the liquid water content during this period (i.e. the soil frozen status).

*9) At the end, the usefulness of P-Band radar is not clear. The results obtain with P-Band radar are limited (Figure 6) but add to the complexity and the length of the manuscript.*

**Response:**

We acknowledged the current P-band radar retrievals are still associated with large uncertainties particularly when ALT is larger than the sensing depth of the P-band radar. We are refining the algorithm (e.g. Chen et al. in press), and working to improve the soil profile parameterization and coupling of the permafrost soil process model with the radar scattering model to reduce the uncertainties in radar retrieval algorithm. As an initial effort towards this, a preliminary analysis on the current radar retrieved results can help us understand the value (and also the limitation) of P-band radar in frozen soil studies. We added the following text to help clarify this:

Page 4, Line 14-23: "Soil dielectric constant is directly associated with the amount of unfrozen water remaining during soil freeze-up, and thus may better define the active layer freezing process comparing with soil temperature. In this study, we investigated the sensitivity of soil dielectric constant to active layer freezing indicated from both in-situ measurements and airborne radar retrievals during the fall transitional period. Longwave (P-band) polarimetric SAR (PolSAR) data with larger penetration depth (~50-60 cm depending on soil moisture content) were acquired from airborne radar acquisitions over northern Alaska in August and October of 2014 and 2015 prior to the NASA Arctic Boreal Vulnerability Experiment (ABoVE) airborne campaign. The airborne radar data were used to characterize spatial variability and seasonal shifts in the near surface (~<10 cm depth) soil dielectric constant associated with the soil F/T transition. These data were used to augment more detailed, but spatially limited in situ soil dielectric measurements used for model parameterization, and to assess the value of longwave radar measurements in frozen soil studies."

*10) 3.1 Model Validation: "model parameterization and validation" would be more appropriate.*

**Response:**

We now move the content in Section 3.1.2 describing the model parameterization into Section 2.4 as suggested by the other reviewer. Therefore the subtitle of Section 3.1 remains unchanged.

*11) p12-28: Have you done an inversion to get that number? Clarify how you calculated the 15%.*

**Response:**

We haven't done an inversion to optimize this threshold, but instead used a trial and error method. We ran multiple model simulations using different values of this parameter acquired from a literature review and ranging from 0-50% (Table 1). We selected the 15% value for this threshold because it produced the smallest RMSE. We tried to clarify this in the manuscript:

Page 10, Line 20-24: "We then tested different soil dielectric thresholds ($\delta$) ranging from 0% to 50% and selected the threshold that produced minimum bias and RMSE between the zero-curtain period determined using in-situ $\varepsilon$ measurements and model simulated unfrozen water content. Using this trial and error method, an optimal threshold of 15% was selected, which produced a mean RMSE of 10.3 days in the simulated zero-curtain from 0.15-0.56 m soil depth in 2016 and 2017 (Fig. 4a)."

*12) p15 – l13: Why higher elevation could impact the relationship? This point should be develop more. Is the elevation impact $\varepsilon 1$ or MODIS SCF, or the soil thermal regime? At what point it is in opposition to the conclusion.*

**Response:**

The reason for a positive correlation between snow cover fraction and the radar retrieved soil dielectric constant difference (August-October) over the IVO transect was not clear. High elevation and large terrain variability as shown in the IVO transect may cause uncertainty in the P-band radar retrievals, due to an increase in terrain roughness and possible errors in the DEM

data used for radar calibration. Additional uncertainties may come from the cloud filtering process in the MODIS SCE data, which used an elevation-based spatial filter (Section 2.1.1). However, the radar algorithm is still under development, and very few in-situ measurements exist in this transect (only one GTN-P site with soil temperature measurements, but with no soil moisture or $\varepsilon$ data). We hope to better quantify the uncertainties of the radar retrieval algorithm in this area in our future work.

*13) Fig S13: I do not see how Fig. S13 shows the effect of snow accumulation. Need more explanation.*

**Response:**

The original Fig. S13 shows the model simulated zero-curtain period at 0.35 m depth for the two years with anomalous late (2007) and early (2015) snow accumulations; the snow accumulations for these two years are shown in the revised Fig. 8 (originally Fig. 7). To avoid confusion, and reduce manuscript length, we replaced the model simulated zero-curtain at 0.25 m shown in the original Fig. 7 with the zero-curtain period simulations at 0.35 m. Therefore, we removed Fig. S13, and updated Fig. 8 in the revised manuscript accordingly. Please refer to the manuscript for more details.

*14) 3.2.2 first paragraph: why considering only two years for this analysis? Does using many years would show that the mean snow onset is correlated to mean zero-curtain period? The results would be much stronger?*

**Response:**

For the analysis, we used the full simulation period except for year 2017 (Fig. 9-10, Fig. S10). The correlation between mean snow onset and mean zero-curtain period for different soil depths was shown in Fig. 10. However, in the first paragraph of section 3.2.2, we first chose two particular years with much earlier and later snow onset than the other years during the simulation period as examples to illustrate how snow cover accumulation in the fall (Fig. 8a-b; 8d-e) affects the length of zero-curtain period.

*15) The model evaluation show some uncertainties in the model. At what point these uncertainties could influence the results and the conclusion?*

**Response:**

One of the major uncertainties in the model simulation should come from the temperature threshold used to define the soil freeze onset based on the shape of the unfrozen water curve, which may vary for different soil conditions (e.g. texture). This could be potentially quantified using a model sensitivity analysis using different unfrozen water curves (different $b$ values in Eq. 11) and temperature thresholds. We added the following sentences to improve clarity:

Page 18, Line 23-26: "A reliable soil dielectric model characterizing the relations between unfrozen water content and $\varepsilon$ for organic soils will help reduce uncertainty in the estimated temperature threshold at soil freeze onset; a model sensitivity analysis using different $b$ values in

the unfrozen water curve (Eq. 11) may also help quantify uncertainties in model simulated zero-curtain period and its regional pattern."

Large uncertainty may exist in the model simulations during ice formation and the liquid water migration process. Better understanding of these processes is needed to reduce this model uncertainty, which is beyond the scope of this paper.

*16) p19 – l5: "information" instead of data?*

**Response:** We removed this sentence.

*17) Figure 1: The magenta and the red stars are very similar.*
    *Figure 5: (b) at 0.35 m?*

**Response:**

- Fig. 2 (originally as Fig. 1): We changed the color scheme of the in-situ datasets.

- Fig. 6 (originally as Fig. 5): We added the depth (0.35 m) in the figure caption.

**Review 2#:**

*1) General comments: "The paper is well written, although parts of it are rather heavy reading (mainly section 2) and could benefit from cutting down some of the text, replaced by perhaps illustrations depicting the analysis process in the form of flowcharts etc. The authors may also consider the usefulness of some of the results, in particular the added value of the P-band SAR observations. Are these really needed, or do they provide only an unnecessary diversion?*

**Response**:

Thank you for the suggestion. We reorganized the Section 2, and added a flowchart to make it easier to follow. We also added a more detailed justification for including the P-band radar retrievals in the revised paper, which is also summarized in our responses to points #3 and #10.

*2) Introduction, P3, lines 6-8: I think citing "high-frequency" instruments here is a bit misleading, as it refers mainly to the FT-ESDR dataset (where high-frequency Ka band data is indeed applied). This statement is not really applicable to scatterometry (e.g. C-band ASCAT) or L-band SMOS and SMAP F/T products. It is a good question how much "high frequencies" indeed tell about soil freezing due to very limited penetration already at C-band in surface vegetation, and the influence of e.g. snow cover at frequencies above Ku band. Of course, the field of view of all of these instruments is very large resulting in coarse-scale products, so that part of the sentence applies (but what about C-band SAR, which are increasingly available from Sentinels and RCM?) I suggest to rewrite this sentence properly acknowledging the various benefits/caveats of different wavelengths. Also add a suitable reference to the scatterometry-based method as well as the SMOS soil F/T product.*

**Response**:

We agreed with the reviewer that we should be specific regarding the soil F/T applications of different satellite sensors and different frequencies. However, current studies still show limited sensitivity of SMAP or SMOS L-band sensors to soils deeper than ~10 cm in boreal forest or wet soil conditions (e.g. Rautiainen et al., 2014). We revised the text to be more specific:

Page 3, Line 6-10: "Moreover, current satellite microwave sensors operating at frequencies ranging from Ka to L-band that provide regional monitoring of surface F/T dynamics are generally less sensitive to deeper soils, e.g. below ~10 cm depth. The soil F/T classification is also constrained by the coarse spatial resolution ($\sim \geq$ 10 km) of passive microwave sensors and scatterometers relative to finer-scale landscape heterogeneity, particularly during seasonal F/T transitions (Naeimi et al., 2012; Rautiainen et al., 2016; Derksen et al., 2017)."

Rautiainen, K., Lemmetyinen, J., Schwank, M., Kontu, A., Ménard, C. B., Mätzler, C., Drusch, M., Wiesmann, A., Ikonen, J. and Pulliainen, J.: Detection of soil freezing from L-band passive microwave observations, Remote Sensing of Environment, 147, 206–218, doi:10.1016/j.rse.2014.03.007, 2014.

*3) Section 2 requires some effort from the reader. It is difficult to follow all the diverse steps required for first gap filling and downscaling MODIS and MERRA-2 snow products, steps required for other reference data, followed by the actual data analysis, and where all these data*

*are finally applied. I understand this is due to the complexity of the analysis, but still it took me two to three read-throughs to finally get a grasp. A flow chart summarizing all of these data preparation steps and the consecutive analysis, indicating clearly where each bit of data is used, could greatly clarify the process. This could be supported by a brief introduction on what is to follow in the beginning of section 2 (following for example the one you have in the beginning of section 3).*

**Response**:

Following the reviewer's suggestion, we added a flowchart (Fig. 1) to better describe the data processing and modeling procedure. To be consistent with the data processing and modeling procedure shown in Fig. 1, we also reorganized Section 2 as follows:

- 2.1 Constructing a fine-resolution snow dataset

- 2.2 The remote sensing driven permafrost soil process model

- 2.3 Model driver datasets and in-situ data

- 2.4 Model parameterization

- 2.5 Regional soil model simulation and analysis

- 2.6 Regional simulation and data analysis

We also added a brief introduction at this section to provide better context and clarity. Please refer to the manuscript for details.

*4) P4 eq. 1. For clarity please define all variables in the equation, even though these may be self-evident (z, t, zs, zb etc.). Furthermore it is not clear how volumetric heat capacity and thermal conductivity are calculated (it is only stated that these "vary" with F/T state, depth etc). Please clarify in the text.*

**Response**:

We now define all of the variables in Eq. 7 (formerly Eq. 1). We also added Eq. 8-10 to better explain how we calculated the volumetric heat capacity and thermal conductivity. Please refer to the manuscript for details.

*5) P7 eq. 7. Again, please give all variables of the equation in the text.*

**Response**:

We now define all of the variables in Eq. 5 (formerly Eq. 7).

*6) P10 line 20. "Mostly focusing" is a bit unclear: can you elaborate? Where are other transects besides DHN used, could be stated here.*

**Response**:

We added the following sentence to explain why we focused on the DHN transect:

Page 12, Line 12-13: "We selected the DHN flight transect as the focus area for the integrated analysis due to the relatively dense network of GTN-P soil temperature and CALM ALT sites in this area relative to other transects (i.e. ATQ and IVO, Fig. 2)."

*7) P11 line 21 "a cutoff threshold of 50%" is not introduced or explained anywhere from what I can see. A reader not familiar with the concept might be highly confused. Please elaborate.*

**Response**:

The cutoff threshold of 50% was used to classify snow or non-snow conditions based on the predicted snow occurrence probability generated by the spatial filter (Fig. 1). This was defined in Eq. 4. We clarified this in the manuscript:

Page 13, Line 11-13: "…which indicates that using a cut-off threshold of 50% for the snow occurrence probability ($P_{cutoff}$, Eq. 4 & Table 1) to classify snow or non-snow conditions works well."

*8) Section 3.1.2, p12 lines 18-26. Might parts of this paragraph be more suited in the methods section? Also, please explain further how thermal conductivity was assumed to "gradually increase". Was the increase linear? This might also give a partial answer to my comment 3, and the explanation might be also better suited for section 2.*

**Response**:

We moved the part in Section 3.1.2 describing the model parameterization to Section 2 "Methods" as suggested, as a separate subsection (Section 2.4: Model parameterization). In Section 2.4, we also explained how we adjusted the soil thermal conductivity:

Page 10, Line 15-19: "The unfrozen soil thermal conductivity within the organic layer was assumed to gradually increase with depth from ~0.5 W m$^{-1}$ K$^{-1}$ in the surface organic soil layer to ~1.2 W m$^{-1}$ K$^{-1}$ at 0.33 m depth for mineral soils, accounting for increases in the soil bulk density (Letts et al., 2000). Similarly, soil porosity was assumed to gradually decease from 0.8 at the surface to ~0.4 in the deeper mineral soil layers. The soil thermal conductivity for frozen conditions can then be determined from Eq. (10)."

We also added Eq. 8-10 to clarify how the model calculated the soil thermal parameterization. The increase in soil thermal conductivity and decrease in soil porosity were linear between the organic and mineral soils. We are working on a more physically based soil parameterization, which can better simulate the smooth transition of soil physical properties from organic to mineral soils, though detailed measurements of these conditions needed for model refinement are lacking in the region.

*9) P13 line 15: "in situ dielectric measurements in frozen soils have significant uncertainties". Why? This should be explained, and references given to support the claim. Also, avoid use of word "significant" unless you are discussing statistical significance.*

**Response**:

We intended to say that there were large uncertainties converting ε to soil moisture (liquid water) at sub-zero temperatures. We revised the sentence for better clarity:

Page 11, line 3-5: "there is a large variability in the relationship between $\varepsilon$ and liquid water content at freezing temperatures due to changes in free and bound water and ice components (Mironov et al., 2010; Naeimi et al., 2012), which can result in large uncertainties in the above estimated thresholds. "

*10) Section 3.2.1. is the inclusion of the P-band airborne SAR observations really necessary? Although the results seem to corroborate your findings, I think these data might be of more interest as a separate paper by itself (provided more analysis can be provided; the data would seem useful for e.g. electromagnetic model exercises from frozen soil). The title of the section is at least strange, change that if nothing else.*

**Response**:

"*The data would seem useful for e.g. electromagnetic model exercises from frozen soil*" – this is a very good suggestion. This topic is also one of our major interests. Due to limited observations provided by the airborne P-band radar (i.e. 2 or 3 snapshots per year), those data, alternatively, should be more useful coupling with a permafrost simulation system to fully explore the sensitivity of P-band radar backscatter to the seasonal thawing and freezing status of the soil active layer (e.g. the spring thaw period, summer maximum thaw period, and the fall freezing period). We are currently working on a modeling framework coupling the permafrost soil model with the radar backscatter model, which can be used for the above sensitivity analysis. As a first step towards this goal, our initial analysis on the current P-band radar retrieved active layer parameters is designed to help us understand the value and limitations of the current radar retrieval algorithm. This activity is part of our larger development framework to develop a more fully coupled radar backscatter and soil dielectric model.

We revised the title of both subsections (3.2.1 and 3.2.2) to be more informative as suggested by the reviewer:

- 3.2.1 Integrated analysis along the DHN airborne flight transect

- 3.2.2 Model sensitivity analysis in Arctic Alaska

*11) P18 line 19 add some suitable references to passive microwave products already here (also e.g. works by Kelly et al.).*

**Response**:

We added the following references of current available global snow water equivalent products using passive microwave:

Page 19, Line 9-10: "Thus, snow properties including snow water equivalent (SWE) may be derived from passive microwave sensors, albeit at relatively coarse spatial scale (Kelly et al., 2003; Armstrong et al., 2005; Takala et al., 2011)."

*12) P18 line 22 "the use of radar is limited:" well this limitation applies to current operational systems as they typically have X-band as the highest frequency. However, QuikSCAT provided already some indication that snow volumetric properties could be captured using radar. Airborne data (e.g. Yueh et al., 2009; King et al., 2018) have provided similar indications (while also revealing limitations). What I am driving at: please be specific about the wavelength and that you are talking about radars currently in space when you make this claim.*

**Response**:

We agree with the reviewer that radar such as Ku band can be sensitive to snow water equivalent that could be quite useful for permafrost and hydrological studies, and its application is not limited to wet snow conditions. However, more work is needed to fully understand the volumetric scattering effects of snow conditions and vegetation canopy on radar backscatter. We revised the text in accordance with reviewer recommendations as follows:

Page 19, Line 11-15: "Compared with passive microwave sensors, active radars or scatterometers such as Ku band are capable of much higher spatial resolution and can be particularly useful for regional snow mapping (Yueh et al., 2009). However, more studies are needed to clarify the multiple scattering effects from snow microstructure variations and contributions from other elements within the sensor footprint including vegetation, soil and open water effects, to ensure accurate retrieval of snow properties (King et al., 2018)."

*13) P18 line 25 "no satellite lidar is currently available:" IceSAT2 just went up, and although no terrestrial snow products are immediately planned, it could be of use also for snow mapping (despite limited coverage). Any comment? Might be good to acknowledge the mission.*

**Response**:

ICESat-2 is potentially useful for mapping regional snow depth both for terrestrial surface and sea ice, though other synergistic data may be needed (e.g. Kwok and Markus, 2018). We revised the sentence as follows:

Page 19, Line 16-17: "while the recently launched ICESat-2 is expected to provide new capabilities of satellite lidar for regional snow mapping (Kwok and Markus, 2018)."

*14) P19: "Radar backscattering measurements:" again I miss definition of the wave-length. A suitably long wavelength would be needed to get meaningful information from subsurface properties, while shorter wavelengths would provide limited information.*

**Response**:

We agreed with the reviewer, that long wavelength (particularly P-band) would be needed for mapping deeper active layer properties. However, radar measurements with shorter wavelengths (e.g. X, C, L-bands) can be also useful for subsurface retrievals, by providing contributing information on surface snow, soil, vegetation conditions, which can be used to reduce the uncertainties on the P-band radar retrievals. However, more sophisticated modeling experiments capable of representing complex landscapes and multi-frequency radar backscatter

characteristics are needed to more fully clarify the value of multi-frequency observations. We revised the text to be more specific as follows:

Page 19, Line 27-29: "Long wavelength radar including P-band (~70 cm) and L-band (~24 cm) is sensitive to surface vegetation structure, soil surface and subsurface dielectric properties…".

Page 20, Line 2-8: "The vegetation canopy also has a large impact on the radar backscatter especially at L-band and shorter wavelengths…Radar measurements with shorter wavelengths (e.g. X, C, L-bands) can be also useful for subsurface retrievals, by providing contributing information on surface snow, soil, and vegetation conditions (Moghaddam et al., 2000; Yueh et al., 2009; King et al., 2018), which can be used to reduce the uncertainties in the longwave (e.g. P-band) radar soil parameter retrievals. However, more sophisticated modelling experiments capable of representing complex landscapes and multi-frequency radar backscatter characteristics are needed to fully clarify the value of multi-frequency observations."

*15) P19: line 9 "similar to all other inversion problems…" Replace 'all' with 'many' or 'typical'? Can't one can find unambiguous inversion problems?*

**Response**:

We replaced "all" with "many". We agree that there are indeed unambiguous inversion problems, as long as there are more independent observations than the target variables to retrieve. However, this is generally not the case for natural systems with more unknown variables than available observations such as the permafrost ecosystem.

*16) Conclusions, p20. At the end you could cite upcoming or planned long-wavelength radar missions, which may be of use for the purpose you cite (NISAR, TanDEM-L, BIOMASS).*

**Response**:

We added additional information in the Conclusions section stating the potential of planned future radar missions in permafrost studies:

Page 20, Line 26-28: "Future satellite P- and L-band radar missions including NISAR, Tandem-L and BIOMASS (Arcioni et al., 2014; Moreira et al., 2015; Rosen et al., 2017) may enable substantial improvements in the way models represent fine-scale soil processes, and thus allow for more accurate predictions of boreal and arctic environmental changes."

*17) Figure 8: add time period for which trends were calculated to figure caption.*

**Response**:

We added the time period (2001-2016) in the caption of Fig. 9 (Originally as Fig.8).

*18) Editorial: 1. P14 line 8 "we then discussed" might sound better as "we then discuss"as the discussion is to follow.*
          *2. Figure 9a: add unit (days) to y-axis.*

**Response**:

We revised the text as suggested, and also updated Figure 10a (originally as Fig. 9a).